# Tracking the Best Expert Privately

Hilal Asi [* 1]   Vinod Raman [* 2]   Aadirupa Saha [* 3]

## Abstract

We design differentially private algorithms for the problem of prediction with expert advice under dynamic regret, also known as tracking the best expert. Our work addresses three natural types of adversaries, stochastic with shifting distributions, oblivious, and adaptive, and designs algorithms with sub-linear regret for all three cases. In particular, under a shifting stochastic adversary where the distribution may shift $S$ times, we provide an $\varepsilon$-differentially private algorithm whose expected dynamic regret is at most $O\left(\sqrt{ST\log(NT)} + \frac{S\log(NT)}{\varepsilon}\right)$, where $T$ and $N$ are the time horizon and number of experts, respectively. For oblivious adversaries, we give a reduction from dynamic regret minimization to static regret minimization, resulting in an upper bound of $O\left(\sqrt{ST\log(NT)} + \frac{ST^{1/3}\log(T/\delta)\log(NT)}{\varepsilon^{2/3}}\right)$ on the expected dynamic regret, where $S$ now denotes the allowable number of switches of the best expert. Finally, similar to static regret, we establish a fundamental separation between oblivious and adaptive adversaries for the dynamic setting: while our algorithms show that sub-linear regret is achievable for oblivious adversaries in the high-privacy regime $\varepsilon \leq \sqrt{S/T}$, we show that any $(\varepsilon, \delta)$-differentially private algorithm must suffer linear dynamic regret under adaptive adversaries for $\varepsilon \leq \sqrt{S/T}$. Finally, to complement this lower bound, we give an $\varepsilon$-differentially private algorithm that attains sub-linear dynamic regret under adaptive adversaries whenever $\varepsilon \gg \sqrt{S/T}$.

*Equal contribution [1]Apple [2]Department of Statistics, University of Michigan, Ann Arbor. Work done while interning at Apple [3]University of Illinios, Chicago. The majority of the work was done while the author was with Apple Research. Correspondence to: Vinod Raman <vkraman@umich.edu>.

*Proceedings of the 42$^{nd}$ International Conference on Machine Learning*, Vancouver, Canada. PMLR 267, 2025. Copyright 2025 by the author(s).

## 1. Introduction

Online learning with experts is a fundamental problem in machine learning, where an online algorithm interacts with an adversary for $T$ rounds (Cesa-Bianchi & Lugosi, 2006). In the general form of the problem with $N$ experts, at each round $t$, the environment chooses a loss vector $\ell_t : [N] \mapsto [0, 1]$, upon which the learner chooses an expert $J_t \in [N]$ from the pool of $N$ experts. In the classical setting of online learning, we measure the loss of the learning algorithm compared to the loss of the 'best-expert' in hindsight, denoted as the (static) regret

$$R_T = \sum_{t=1}^{T} \ell_t(J_t) - \min_{j^\star \in [N]} \sum_{t=1}^{T} \ell_t(j^\star).$$

However, comparing against a single fixed expert can often be unrealistic in practical applications. Even the best fixed expert may perform poorly on average over the entire loss sequence, especially when loss sequences dynamically change over time or undergo significant distributional shifts, as is common in stochastic settings. This limitation motivates the concept of dynamic regret (Herbster & Warmuth, 1998; Wei et al., 2016), which provides a more flexible and robust benchmark. Unlike static regret, which evaluates against the best fixed expert, dynamic regret compares against a sequence of changing experts, enabling the model to adapt to evolving environments. In particular, the dynamic regret is

$$DR_T = \sum_{t=1}^{T} \ell_t(J_t) - \min_{j_1^\star, \ldots, j_T^\star \in [N]} \sum_{t=1}^{T} \ell_t(j_t^\star),$$

subject to the constraint that the sequence $\{j_t^\star\}$ does not switch too often. Intuitively, dynamic regret measures how well the algorithm competes against the best possible sequence of decisions that could adapt to changes, constrained by a limited number of switches between experts. This notion is particularly relevant for real-world applications like financial markets, where optimal strategies vary with market conditions, or recommendation systems, where user preferences evolve over time.

By inspecting definitions, it is clear that minimizing dynamic regret is *harder* than static regret. This difficulty manifests even if one measures dynamic regret against sequences of experts with a *single* switch. As a result, exist-

ing algorithms for minimizing dynamic regret modify the standard Multiplicative Weights Algorithm (Littlestone & Warmuth, 1994) to explicitly account for the fact that they are being evaluated against sequences of experts that switch.

While dynamic regret in online learning offers a more practical approach to modeling non-stationary environments, its applicability in sensitive real-world scenarios often requires additional considerations, particularly around privacy. In many online learning problems, the loss functions used to guide expert selection are derived from sensitive data, such as user interactions, medical information, or financial records. Ensuring that the learning process does not inadvertently reveal private details about the data is crucial for maintaining trust and complying with legal and ethical standards. This challenge motivates the integration of differential privacy into online learning from experts in the dynamic setting.

However, existing work on private online learning (Jain et al., 2012a; Smith & Thakurta, 2013; Jain & Thakurta, 2014a; Agarwal & Singh, 2017a; Asi et al., 2023b; 2024) is limited to the static setting. As a result, existing privacy-preserving algorithms struggle to adapt to non-stationary environments, where the optimal expert may shift over time, leading to suboptimal performance.

Our work addresses this gap by initiating the study of private online prediction from experts in the dynamic setting. We formally define this problem and study it with respect to three natural types of adversaries. We develop new algorithms and lower bounds for each of these adversaries, demonstrating the near-optimality of our algorithms in several settings, and the hardness of the dynamic setting compared to the well-studied static setting.

### 1.1. Our Contributions

In this work, we initiate and systematically study the problem of tracking the best expert privately through the lens of online prediction with dynamic regret guarantees. We present a comprehensive study of the problem for three different types of adversaries: 1. Shifting stochastic adversaries where the losses are sampled from distributions that may shift over time, 2. Oblivious adversaries which choose the loss functions before the interaction with the algorithm, and 3. Adaptive adversaries, the most powerful type of adversary, which can choose their loss functions as a function of the interaction with the learning algorithm. We highlight the following key results:

**Shifting stochastic adversaries.** We design an $\varepsilon$-differentially private algorithm with an expected dynamic regret bound of $O\left(\sqrt{ST\log(TN)} + \frac{S\log(TN)}{\varepsilon}\right)$, where $T$, $N$, and $S$ represent the time horizon, number of experts,

and number of distribution shifts, respectively. We also give a lower bound of $\Omega(\sqrt{ST\log(N)} + S\log(N/S)/\varepsilon)$ for this setting, demonstrating the near-optimality of this algorithm. Key to our algorithm is the sparse-vector-technique which we deploy in order to identify a new shift in the distribution without paying a large cost in privacy.

**Oblivious adversaries.** We develop a new algorithm for the oblivious setting through a reduction from private online prediction in the dynamic setting to the static setting. Applying this reduction with existing algorithms for private prediction in the static setting (Asi et al., 2023b), we obtain an upper bound of $O\left(\sqrt{TS\log(NT)} + \frac{ST^{1/3}\log(T/\delta)\log(NT)}{\varepsilon^{2/3}}\right)$ on the expected dynamic regret.

**Adaptive adversaries.** We establish a separation between oblivious and adaptive adversaries in the dynamic setting. To this end, we show that any $(\varepsilon, \delta)$-differentially private algorithm must suffer linear dynamic regret $\Omega(T)$ under adaptive adversaries for $\varepsilon \leq \sqrt{S/T}$. In contrast, our upper bounds for oblivious adversaries show that sub-linear regret is still possible for $\varepsilon \leq \sqrt{S/T}$. Finally, we provide a new algorithm that obtain sub-linear regret for $\varepsilon \gg \sqrt{S/T}$. This establishes $\varepsilon \approx \sqrt{S/T}$ as a critical sharp threshold for learning under adaptive adversaries in the dynamic setting, where learning becomes infeasible for $\varepsilon \leq \sqrt{S/T}$ but is attainable for larger values of $\varepsilon$.

### 1.2. Related Works

**Private Online Learning and Prediction with Expert Advice.** Differentially private online learning was first studied by Dwork et al. (2010a) in the context of continual observations. Jain et al. (2012b) extend these results to online convex programming by using gradient-based algorithms to achieve differential privacy. Following this work, Guha Thakurta & Smith (2013) privatize the Follow-the-Approximate-Leader template to obtain sharper guarantees for online convex optimization. For prediction with expert advice, Dwork et al. (2014) and Jain & Thakurta (2014b) give private online learning algorithms with regret bounds of $O\left(\frac{\sqrt{T\log(N)}}{\varepsilon}\right)$. More recently, (Agarwal & Singh, 2017b) study private online linear optimization and achieve regret bounds that scale like $O(\sqrt{T}) + O(\frac{1}{\varepsilon})$. Using this result, they show that for the setting of prediction with expert advice, it is possible to obtain a regret bound that scales like $O\left(\sqrt{T\log(N)} + \frac{N\log(N)\log^2 T}{\varepsilon}\right)$, improving upon the work by Dwork et al. (2014) and Jain & Thakurta (2014b). For large $N$, this upper bound was further improved to $O\left(\sqrt{T\log(N)} + \frac{T^{1/3}\log(N)}{\varepsilon}\right)$ and $O\left(\sqrt{T\log(N)} + \frac{T^{1/3}\log(N)}{\varepsilon^{2/3}}\right)$ by Asi et al. (2023b) and

*Table 1.* Summary of our bounds on the dynamic regret for different settings of the adversary. We omit logarithmic factors in $T$ and $1/\delta$.

| | **Upper Bound** | **Lower Bound** |
|---|---|---|
| **Shifting Stochastic** | $\sqrt{ST \log N} + \frac{S \log N}{\varepsilon}$ | $\sqrt{ST \log N} + \frac{S \log(N/S)}{\varepsilon}$ |
| **Oblivious** | $\sqrt{ST \log N} + \frac{ST^{1/3} \log N}{\varepsilon^{2/3}}$ | $\sqrt{ST \log N} + \frac{S \log(N/S)}{\varepsilon}$ |
| **Adaptive** | $\frac{\sqrt{ST \log^{1.5} N}}{\varepsilon} + \frac{S \log N}{\varepsilon}$ | $\sqrt{ST \log N} + \frac{S}{\left(\varepsilon \log \frac{T}{S+1}\right)^2}$ |

Asi et al. (2024) respectively, under an oblivious adversary. Recent work also study private prediction with expert advice in the realizable setting where there is a zero-loss expert (Asi et al., 2023a).

Asi et al. (2023b) also study private prediction with expert advice under stochastic and adaptive adversaries. Under a stochastic adversary, they reduce private online learning to private offline learning and give an $(\varepsilon, \delta)$-differentially private online learning algorithm with expected regret $O(\sqrt{T \log(N)} + \frac{\log N}{\varepsilon})$. Under adaptive adversaries, (Asi et al., 2023b) prove a lower bound – any $(\varepsilon, \delta)$-differentially private online algorithm with $\varepsilon \leq \frac{1}{\sqrt{T}}$ cannot achieve sublinear regret under an adaptive adversary. This result established a separation between the achievable regret bounds under oblivious and adaptive adversaries.

**Non-private Dynamic and Adaptive Regret Minimization.** In the context of prediction with expert advice, dynamic regret minimization is also known as tracking the best expert (Littlestone & Warmuth, 1994; Herbster & Warmuth, 1998; Gyorgy et al., 2012; Bousquet & Warmuth, 2002; Vovk, 1997). This setting was first introduced by Herbster & Warmuth (1998; 2001), who noted that static regret is only meaningful for stationary environments. Following this work, there has been significant interest in obtaining dynamic regret bounds for various settings. For example, Wei et al. (2016) study dynamic regret bounds in non-stationary stochastic environments, while Zinkevich (2003); Hall & Willett (2013) have studied dynamic regret for online optimization problems. Other works have also focused on obtaining first- and second-order dynamic regret bounds (Zhang et al., 2018; Lu & Zhang, 2019) and obtaining guarantees for dynamic regret for stochastic and oblivious adversaries simultaneously (Luo & Schapire, 2015). Most relevant to this paper is the work by Lu & Zhang (2019), who provide a simple modification to the standard Multiplicative Weights Algorithm (Littlestone & Warmuth, 1994) that obtains near optimal dynamic regret under an oblivious adversary.

A closely related notion to dynamic regret is adaptive regret (Littlestone & Warmuth, 1994; Hazan & Seshadhri, 2007). Here, the goal is to obtain sublinear regret within every contiguous sub-interval of the time horizon. Several works have established deep connections between adaptive

and dynamic regret for the setting of prediction with expert advice (Adamskiy et al., 2012; Cesa-Bianchi et al., 2012; Daniely et al., 2015). In fact, for prediction with expert advice, it is known that the dynamic regret can be upper bounded by the adaptive regret, and hence adaptive regret minimization is sufficient for dynamic regret minimization (Luo & Schapire, 2015). In this paper, we use this connection between adaptive and dynamic regret minimization to derive bounds on the dynamic regret under stochastic adversaries under privacy constraints.

## 2. Preliminaries

Let $N \in \mathbb{N}$ denote the number of experts and $\ell : [N] \mapsto [0, 1]$ denote an arbitrary loss function that maps an expert to a bounded loss. For an abstract sequence $z_1, \ldots, z_n$, we abbreviate it as $z_{1:n}$. For a measurable space $(\mathcal{X}, \sigma(\mathcal{X}))$, we let $\Delta \mathcal{X}$ denote the set of all probability measures on $\mathcal{X}$. For $N \in \mathbb{N}$, we also let $\Delta_N$ denote the set of all distributions over $\{1, \ldots, N\}$. We let $\mathsf{Laplace}(\lambda)$ denote the Laplace distribution with mean zero and scale $\lambda$ such that its probability density function is $f_\lambda(x) = \frac{1}{2\lambda} \exp\left(\frac{-|x|}{\lambda}\right)$. Finally, we let $[N] := \{1, \ldots, N\}$ for $N \in \mathbb{N}$.

### 2.1. Prediction with Expert Advice and Static Regret

In the classical problem of online prediction with expert advice, a learning algorithm $\mathcal{A}$ plays a sequential game against an adversary over $T \in \mathbb{N}$ rounds. In full generality, the adversary first picks a sequence of functions $f_1, f_2, \ldots, f_T$ such that $f_t : [N] \times [N]^{t-1} \to [0, 1]$ for all $t \in [T]$. Then, in each round $t \in [T]$, the learner, using the history of the game, selects (potentially randomly) expert $J_t \in [N]$. Finally, the adversary reveals the loss function $\ell_t := f_t(\cdot, J_{1:t-1})$ and the learner suffers the loss $\ell_t(J_t)$. The goal of the learner is to adaptively select experts $J_1, \ldots, J_T \in [N]$ such as to minimize its *expected (static) regret*

$$
R_{\mathcal{A}}(f_{1:T}, N) := \mathbb{E}_{\mathcal{A}}\left[ \sum_{t=1}^{T} f_t(J_t, J_{1:t-1}) \right.
$$
$$
\left. - \min_{j^\star \in [N]} \sum_{t=1}^{T} f_t(j^\star, J_{1:t-1}) \right],
$$

where the expectation is taken only with respect to the randomness of the learning algorithm.

## 2.2. Dynamic Regret

Motivated by concerns of distribution shift, there has been significant interest in minimizing a stronger notion of expected regret termed expected *dynamic* regret (Herbster & Warmuth, 1998). Unlike expected (static) regret, where the goal is to compete against the best fixed expert in hindsight, dynamic regret measures the performance of the player against a comparison *sequence* of experts $j_{1:T} \in [N]^T$. To make the problem tractable, we constrain the comparison sequence of experts to have at most $S$ switches, where $S \in \mathbb{N}$ is known to the player before the game begins. Namely, a sequence of $T$ experts $j_{1:T} \in [N]^T$ has at most $S$ switches if $\sum_{t=1}^{T-1} \mathbb{1}\{j_{t+1} \neq j_t\} \leq S$. Then,

$$\mathcal{C}(T, S) := \left\{ j_{1:T} \in [N]^T : \sum_{t=1}^{T-1} \mathbb{1}\{j_{t+1} \neq j_t\} \leq S \right\},$$

is the set of all $T$-length expert sequences with at most $S$ switches. For a sequence of functions $f_1, f_2, \ldots, f_T$, we can now define the expected dynamic regret for an algorithm $\mathcal{A}$ by comparing its cumulative loss to that of the best fixed *sequence* of experts in $\mathcal{C}(T, S)$:

$$\mathrm{DR}_{\mathcal{A}}(f_{1:T}, N, S) := \mathbb{E}_{\mathcal{A}} \left[ \sum_{t=1}^{T} f_t(J_t, J_{1:t-1}) \right. $$
$$\left. - \min_{j_{1:T}^{\star} \in \mathcal{C}(T, S)} \sum_{t=1}^{T} f_t(j_t^{\star}, J_{1:t-1}) \right].$$

We make a few remarks about the definition of dynamic regret. First, note that $\mathrm{R}_{\mathcal{A}}(f_{1:T}, N) = \mathrm{DR}_{\mathcal{A}}(f_{1:T}, N, 0)$. Second, $\mathrm{DR}_{\mathcal{A}}(f_{1:T}, N, S_1) \leq \mathrm{DR}_{\mathcal{A}}(f_{1:T}, N, S_2)$ for $S_1 \leq S_2$, meaning that as $S$ gets larger, the set of comparison sequence of experts $\mathcal{C}(T, S)$ gets larger, making minimizing dynamic regret *harder*. Lastly, we stress that while dynamic regret restricts the number of switches in the comparison sequence of experts, the player is *not* restricted in the number of switches it can make. This is crucial for being able to obtain sublinear expected dynamic regret.

By placing restrictions on how $f_1, f_2, \ldots, f_T$ can be chosen, one gets different types of adversaries leading to different definitions of *worst-case* expected dynamic regret. In this paper, we consider three adversaries: (1) shifting stochastic, (2) oblivious, and (3) adaptive.

The strongest of the three is the *adaptive* adversary. For an adaptive adversary, no restrictions are placed - the adversary can pick *any* sequence of functions $f_1, f_2, \ldots, f_T$ leading to the worst-case expected dynamic regret being defined as

$$\mathrm{DR}_{\mathcal{A}}^{\mathrm{adap}}(T, N, S) := \sup_{f_1, \ldots, f_T} \mathrm{DR}_{\mathcal{A}}(f_{1:T}, N, S).$$

A weakening of the adaptive adversary is an *oblivious* adversary. This adversary must first pick a sequence of loss vectors $\ell_1, \ldots, \ell_T$ independently of $J_{1:T}$ and then construct the sequence of functions $f_{\ell_1}, f_{\ell_2}, \ldots, f_{\ell_T}$ such that $f_{\ell_t}(j_t, j_{1:t-1}) := \ell_t(j_t)$. We define the worst-case expected regret under an oblivious adversary as

$$\mathrm{DR}_{\mathcal{A}}^{\mathrm{obl}}(T, N, S) := \sup_{\ell_1, \ldots, \ell_T} \mathrm{DR}_{\mathcal{A}}(f_{\ell_1}, \ldots, f_{\ell_T}, N, S).$$

Finally, the *shifting stochastic* adversary is the weakest of the three. Here, the adversary must first pick a sequence of $S$ distributions $\mathcal{D}_1, \ldots, \mathcal{D}_S \in \Delta([0, 1]^N)$ and a sequence of $S - 1$ time points $t_1, \ldots, t_{S-1} \in [T - 1]$. The adversary draws loss functions $\ell_1, \ldots, \ell_T$ such that $\ell_t \sim \mathcal{D}_s$ iff $t \in [t_s, t_{s+1})$ and constructs functions $f_{\ell_1}, f_{\ell_2}, \ldots, f_{\ell_T}$. Abusing some notation by omitting the dependence on $t_{1:S-1}$, the worst-case expected regret under a stochastic adversary is

$$\mathrm{DR}_{\mathcal{A}}^{\mathrm{stoc}}(T, N, S) := \sup_{\mathcal{D}_{1:S}} \mathbb{E}_{\ell_{1:T} \sim \mathcal{D}_{1:S}} \left[ \mathrm{DR}_{\mathcal{A}}(f_{\ell_1}, \ldots, f_{\ell_T}, N, S) \right].$$

Note that in the definition of $\mathrm{DR}_{\mathcal{A}}^{\mathrm{stoc}}(T, N, S)$, the same $S$ is used to constrain both the number of distributions that the adversary can pick and the number of switches in the comparison sequence of experts.

Analogous versions of *worst-case* expected (static) regret under adaptive, oblivious, and shifting stochastic adversaries follow by placing the same restrictions on how $f_1, \ldots, f_T$ can be chosen. As an example, the worst-case expected (static) regret under an adaptive adversary will be written as $\mathrm{R}_{\mathcal{A}}^{\mathrm{adap}}(T, N) = \sup_{f_1, \ldots, f_T} \mathrm{R}_{\mathcal{A}}(f_{1:T}, N)$. Without privacy concerns, the worst-case expected regret is $\Theta(\sqrt{T \log N})$ for all three types of adversaries and can be obtained by running a single algorithm, the Multiplicative Weights Algorithm (MWA) (Littlestone & Warmuth, 1994). Likewise, the worst-case expected *dynamic* regret under stochastic, oblivious, and adaptive adversaries is also known to be $\Theta(\sqrt{TS \log N})$, and achieved by *modifying* MWA to ensure that the probability of playing any expert never drops too low (where "low" depends on $S$ and $T$) (Herbster & Warmuth, 1998; Wei et al., 2016).

## 2.3. Differential Privacy

We adopt the notion of differential privacy for prediction with expert advice from Asi et al. (2023b). Consider an abstract space $\mathcal{Z}$ and consider a function $\ell : [N] \times \mathcal{Z} \to [0, 1]$. Every $z \in \mathcal{Z}$ now induces a loss function $\ell(\cdot, z) \in [0, 1]^N$. Accordingly, stochastic, oblivious, and adaptive adversaries can be equivalently defined in terms of picking $z_1, \ldots, z_T \in \mathcal{Z}^T$ and a function $\ell : [N] \times \mathcal{Z} \to [0, 1]$. For completeness sake, we make this explicit below.

A stochastic adversary under dynamic regret now picks a sequence of $S$ distributions $\mathcal{D}_1, \ldots, \mathcal{D}_S \in \mathcal{Z}$, a sequence of times points $t_1, \ldots, t_{S-1}$, and a function $\ell : [N] \times \mathcal{Z} \to [0, 1]$. The loss function at time point $t \in [t_s, t_{s+})$ is obtained by sampling $z_t \in \mathcal{D}_s$ and outputting $\ell_t = \ell(\cdot, z_t)$. An oblivious adversary selects a sequence $z_1, \ldots, z_T \in \mathcal{Z}^T$ and a function $\ell : [N] \times \mathcal{Z} \to [0, 1]$ before the game begins. The loss function at time $t \in [T]$ is then defined as $\ell_t := \ell(\cdot, z_t)$. Finally an adaptive adversary picks a sequence $z_1, \ldots, z_T \in \mathcal{Z}^T$ and a sequence of function $\ell_t : [N] \times [N]^{t-1} \times \mathcal{Z} \to [0, 1]$. The loss function at time $t \in [T]$, is then defined by $\ell_t(\cdot, J_{1:t-1}, z_t)$, where $J_{1:t-1}$ are the random variable representing the actions of the player. Note that give an input $z_{1:T}$, stochastic and oblivious adversaries are fully parameterized by a function $\ell : [N] \times \mathcal{Z} \to [0, 1]$, while adaptive adversaries are parameterized by a sequence of functions $\ell_1, \ldots, \ell_T$ such that $\ell_t : [N] \times [N]^{t-1} \times \mathcal{Z} \to [0, 1]$.

With this equivalent representation in mind, we are now ready to define our notion of differential privacy. A dataset is as sequence of elements $z_1, \ldots, z_T$. Two datasets, $z_{1:T}$ and $z'_{1:T}$, are neighboring if they differ exactly at one time point $t' \in [T]$. Let $\mathcal{A} \circ \text{Adv}(z_{1:T}) = J_1, \ldots, J_T$ be the sequence of random variables denoting the experts played by $\mathcal{A}$ when interacting with the adversary $\text{Adv}$ that is given inputs $z_{1:T}$.

**Definition 2.1** (Adaptive Differential Privacy)**.** A randomized algorithm $\mathcal{A}$ is $(\varepsilon, \delta)$-differentially private against adaptive adversaries, if for all neighboring data sets $z_{1:T}, z'_{1:T} \in \mathcal{Z}^T$, any potentially adaptive adversary $\text{Adv}$, and all events $E \subseteq [N]^T$, we have that

$$\mathbb{P}\left[\mathcal{A} \circ \text{Adv}(z_{1:T}) \in E\right] \le e^{\varepsilon} \mathbb{P}\left[\mathcal{A} \circ \text{Adv}(z'_{1:T}) \in E\right] + \delta.$$

If $\delta = 0$, we say that $\mathcal{A}$ is $\varepsilon$-differentially private.

The following mechanisms will also be useful building blocks to several of our algorithms.

**Laplace Mechanism.** Let $\mathcal{X}$ be an arbitrary set and $n \in \mathbb{N}$. Suppose $f : \mathcal{X}^n \to \mathbb{R}$ is a query with sensitivity $\Delta$ (i.e. for all pairs of datasets $x_{1:n}, x'_{1:n} \in \mathcal{X}^n$ that differ in exactly one index, we have that $|f(x_{1:n}) - f(x'_{1:n})| \le \Delta$). Then, for every $\varepsilon$, the Laplace mechanism $\mathcal{M} : \mathcal{X}^n \to \mathbb{R}$ is defined as $\mathcal{M}(x_{1:n}) = f(x_{1:n}) + Z$, where $Z \sim \text{Lap}(\frac{\Delta}{\varepsilon})$.

**Lemma 2.2** ((Dwork & Roth, 2014), Theorem 3.6)**.** *The Laplace Mechanism is $\varepsilon$-differentially private.*

**Report-Noisy-Max Mechanism.** The report-noisy-max mechanism is a differentially private algorithm that aims to select the item with the highest count. More specifically, given an input dataset $x_{1:n} \in \mathcal{X}^n$ and $K$ count queries $c_1, \cdots, c_K : \mathcal{X}^n \to \mathbb{R}$ that are 1-sensitive, report-noisy-

max returns

$$j = \arg\max_{i \in [K]} c_i(x_{1:n}) + Z_i, \text{ where } Z_i \sim \text{Laplace}(2/\varepsilon).$$

**Lemma 2.3** ((Dwork & Roth, 2014), claim 3.9)**.** *The report-noisy-max algorithm is $\varepsilon$-differentially private.*

**Sparse vector technique.** We recall the sparse-vector-technique (Dwork & Roth, 2014). Given an input $x_{1:n} \in \mathcal{X}^n$, the algorithm takes a stream of queries $q_1, q_2, \ldots, q_T : \mathcal{X}^n \to \mathbb{R}$ in an online manner and aims to identify queries whose value is above zero. We assume that each $q_i$ is 1-sensitive, that is, $|q_i(x_{1:n}) - q_i(x'_{1:n})| \le 1$ for neighboring datasets $x_{1:n}, x'_{1:n}$ that differ in a single element. We have the following guarantee.

**Lemma 2.4** ((Dwork & Roth, 2014), Theorem 3.24)**.** *Let $x_{1:n} \in \mathcal{X}^n$ be an input dataset. For $\beta > 0$, there is an $\varepsilon$-differentially private algorithm (AboveThreshold) that halts at time $k \in [T+1]$ such that for $\alpha = \frac{8(\log T + \log(2/\beta))}{\varepsilon}$ with probability at least $1 - \beta$,*

- *For all $t < k$, $q_i(x_{1:n}) \le \alpha$;*

- *$q_k(x_{1:n}) \ge -\alpha$ or $k = T + 1$.*

To facilitate the notation for using AboveThreshold in our algorithms, we assume that it has the following components:

1. InitializeSparseVec$(\varepsilon, \beta)$: initializes a new instance of AboveThreshold with privacy parameter $\varepsilon$, and failure probability parameter $\beta$. This returns an instance (data structure) $Q$ that supports the following test-above-threshold function.

2. $Q$.TestAboThr$(q)$: tests if the query $q$ is above threshold. In that case, the algorithm stops and does not accept more queries.

# 3. SVT-based Algorithm for Stochastic Adversaries

We begin our algorithmic contribution by studying the stochastic setting, where we develop an SVT based algorithm that obtains $\sqrt{ST} + S/\varepsilon$ dynamic regret against stochastic adversaries. The starting point of our algorithm is lazy algorithm of (Asi et al., 2023b) for private prediction from experts with static regret. We show in Section 3.1 that this algorithm obtains near-optimal adaptive regret (Hazan & Seshadhri, 2007), that is, it obtains regret $\sqrt{w}$ for any sub-interval of size $w$ for a stochastic adversary. Then, we present our main algorithm in Section 3.2, which obtains near-optimal dynamic regret against stochastic adversaries.

## 3.1. Optimal Adaptive Regret for Stationary Environment

In this section, we consider the simple stochastic setting where all losses are samples for a fixed distribution $P$. We present that a version of the existing algorithm of (Asi et al., 2023b) obtains a stronger guarantee than the original paper proved: it obtains near-optimal adaptive regret for stochastic adversaries, meaning that it obtains $\sqrt{w}$ regret for any sub-interval of size $w$.

---

**Algorithm 1** Limited Updates for Online Optimization with Stochastic Adversaries

---

**Require:** Privacy parameter $\varepsilon$
1: Set $j_0 \in [N]$
2: **for** $t = 1$ to $T$ **do**
3:     **if** $t = 2^\ell$ for some integer $\ell \geq 1$ **then**
4:       Run report-noisy-max procedure to get

$$j_t = \arg\min_{j \in [N]} \sum_{i=t/2}^{t-1} \ell_i(j) + Z_t(j),$$

$$\text{where } Z_t(j) \sim \mathsf{Laplace}(2/\varepsilon)$$

5:     **else**
6:       Let $j_t = j_{t-1}$
7:     **end if**
8:     Receive $\ell_t : [N] \to [0,1]$.
9:     Pay cost $\ell_t(j_t)$
10: **end for**

---

The following theorem states the adaptive regret guarantees of Algorithm 1. We defer the proof to Appendix B.1.

**Theorem 3.1.** *Let $\ell_1, \ldots, \ell_T : [N] \to [0,1]$ be sampled i.i.d. from a distribution $P$. Then, for any $t \in [T]$ and $w \in [T-t]$, Algorithm 1 is $\varepsilon$-differentially private and has with probability $1 - \beta$,*

$$\sum_{i=t}^{t+w} \ell_i(j_i) - \min_{j \in [N]} \sum_{i=t}^{t+w} \ell_i(j) \leq \frac{16 \log(NT/\beta) \log(T)}{\varepsilon}$$
$$+ 9\sqrt{w \log(TN/\beta)}.$$

## 3.2. Optimal Dynamic Regret for Shifting Stochastic Adversaries

In this section, we develop our main algorithm for the stochastic setting. Our algorithm is based on iteratively running the algorithm for the stationary setting (Algorithm 1). Moreover, to adapt to shifting distributions, our algorithm uses the sparse-vector-technique to test whether the underlying distribution of the losses has changes. To this end, we use SVT to test whether the regret of the internal algorithm is too large, indicating a shift in the distribution. We present the full details in Algorithm 2.

---

**Algorithm 2** SVT-based algorithm

---

**Require:** Privacy parameter $\varepsilon$, failure probability $\beta$
1: $t_1 = 1$, $i = 1$
2: Start new instance of Algorithm 1 from $t_i$ with privacy parameter $\varepsilon/2$
3: $Q = \mathsf{InitializeSparseVec}(\varepsilon/2, \beta/T)$
4: **while** $t < T$ **do**
5:     Receive new loss $\ell_t$
6:     Use Algorithm 1 to play $j_t$
7:     Define $\alpha = \frac{16(2\log T + \log(2/\beta))}{\varepsilon}$ and $\mathsf{Reg}_w := \frac{16 \log(NT/\beta) \log(T)}{\varepsilon} + 9\sqrt{w \log(TN/\beta)}$
8:     For each $w \leq t - t_i$, define query

$$q_w^t := \sum_{i=t-w}^{t} \ell_i(j_i) - \min_{j \in [N]} \sum_{i=t-w}^{t} \ell_i(j) - \mathsf{Reg}_w - \alpha - 1$$

9:     **if** Q.TestAboThr$(q_w^t)$ is true for some $w$ **then**
10:       $i \to i + 1$
11:       $t_i = t$
12:       Go to line 2 and restart a new instance of Algorithm 1
13:     **end if**
14: **end while**

---

The following theorem summarizes the dynamic regret guarantees of Algorithm 2. We defer the proof to Appendix B.2.

**Theorem 3.2** (Upper bounds for Expected Dynamic Regret for Shifting Stochastic Adversaries)**.** *Let $\mathcal{A}$ denote Algorithm 2 when run with $\varepsilon$ and $\beta = 1/T$. Then algorithm $\mathcal{A}$ is $\varepsilon$-differentially private and has expected dynamic regret $\mathrm{DR}_{\mathcal{A}}^{\mathsf{stoc}}(T, N, S)$ upper bounded by*

$$O\left(\sqrt{ST \log(NT)} + \frac{S \log(NT) \log(T)}{\varepsilon}\right).$$

*Proof.* (sketch) The privacy proof follows directly from the guarantees of SVT mechanism and Algorithm 1, as each user is used in the instantiation of both Algorithm 1 and SVT with parameters $\varepsilon/2$.

The utility proof follows from two main ingredients, which we prove in Lemma B.3 and Lemma B.4. The first shows that if the distribution shifts at most $S$ times, then SVT will return true at most $S$ times with high probability. The second result shows that as long as SVT has not restarted the instantiation of the internal algorithm, its regret in any sub-interval (adaptive regret) will be small. Building on these two lemmas, we can prove an upper bound on the dynamic regret (see Appendix B.2 for full details). $\square$

**Lower bound for shifting adversary.** We can extend the lower bound of the static setting (Asi et al., 2023b) to our dynamic setting. Indeed, the static setting has a lower bound

of $\log(N)/\varepsilon$ on the expected regret. We can construct an adversary which splits the rounds to $S$ phases, where in each phase it uses the static lower bound over a disjoint subset of the experts of size $N/S$. Given the independence of these phases, this implies that the regret in each phase is lower bounded by $\log(N/S)/\varepsilon$. Summing over all $S$ phases, we get that the dynamic regret is lower bounded by $S\log(N/S)/\varepsilon$. Finally, note that in the most common setting of parameters where $S \leq T \leq N^{1-\rho}$ for some constant $\rho > 0$, this lower bound becomes $\Omega(S\log(N)/\varepsilon)$, matching our upper bound.

## 4. Upper bounds for Oblivious Adversaries

To obtain our upper bounds on expected dynamic regret against oblivious adversaries, we reduce private dynamic regret minimization to private static regret minimization. That is, our main result is a conversion of a private online learning algorithm minimizing static regret to a private online learning algorithm minimizing dynamic regret. By doing so, we are able to leverage the recent results by Asi et al. (2024), who obtain the best-known expected (static) regret guarantees for private online learning under oblivious adversaries.

**Theorem 4.1** (Private Static Regret $\implies$ Private Dynamic Regret). *Let $\varepsilon, \delta \in (0, 1)$. Suppose there exists an $(\varepsilon, \delta)$-differentially private algorithm $\mathcal{A}$ whose worst-case expected regret under an oblivious adversary is at most $\mathrm{R}_{\mathcal{A}}^{\mathrm{obl}}(T, N)$. Then, there exists an $(\varepsilon, \delta)$-differentially private algorithm $\mathcal{B}$ such that $\mathrm{DR}_{\mathcal{B}}^{\mathrm{obl}}(T, N, S) \leq \mathrm{R}_{\mathcal{A}}^{\mathrm{obl}}(T, (NT)^{2S})$.*

*Proof.* Let $\varepsilon, \delta \in (0, 1)$. Fix the time horizon $T \in \mathbb{N}$, the number of experts $N \in \mathbb{N}$, and the number of switches $S \in \mathbb{N}$. Suppose there exists an $(\varepsilon, \delta)$-differentially private algorithm $\mathcal{A}$ whose worst-case expected regret under an oblivious adversary is at most $\mathrm{R}_{\mathcal{A}}^{\mathrm{obl}}(T, N)$.

Consider the following algorithm $\mathcal{B}$. Let $[T]_{\leq c}$ be the set of all strictly increasing tuples of size at most $c$. Before the game beings, $\mathcal{B}$ first constructs the class of meta-experts $\mathcal{E}$ such that

$$\mathcal{E} = \bigcup_{c=0}^{S} \left\{ e_{t_{1:c}, j_{1:c+1}} : t_{1:c} \in [T]_{\leq c}, j_{1:c+1} \in [N]^{c+1} \right\}$$

where the expert $e_{t_{1:c}, j_{1:c+1}} : [T] \to [N]$ plays expert $j_i$ from time point $t_{i-1}$ to $t_i$ for every $i \in [c+1]$, where $t_0 = 1$ and $t_{c+1} = T$. Then, $\mathcal{B}$ initializes $\mathcal{A}$ with the set of meta experts $\mathcal{E}$. In each round $t \in [T]$, $\mathcal{B}$ queries $\mathcal{A}$, receives a (potentially random) meta-expert $E_t \in \mathcal{E}$ from $\mathcal{A}$, and plays the expert $J_t \in [N]$ played by the meta-expert $E_t$ on round $t$. That is, $J_t := E_t(t)$. After observing the true loss vector $\ell_t : [N] \to [0, 1]$, $\mathcal{B}$ computes a meta-loss vector

$\tilde{\ell}_t : \mathcal{E} \to [0, 1]$ such that $\tilde{\ell}_t(e) := \ell_t(e(t))$ for all $e \in \mathcal{E}$ and passes $\tilde{\ell}_t$ to $\mathcal{A}$, which then updates itself. We claim that: (1) $\mathcal{B}'s$ expected dynamic regret is at most $\mathrm{R}_{\mathcal{A}}^{\mathrm{obl}}(T, (NT)^{2S})$ and (2) $\mathcal{B}$ is $(\varepsilon, \delta)$-differentially private.

We start by proving Claim (1). Observe that

$$|\mathcal{E}| = \sum_{c=0}^{S} \binom{T}{c} N^{c+1} \leq N^{S+1} \sum_{c=0}^{S} \binom{T}{c} \leq (NT)^{2S}.$$

Therefore, by the guarantees of $\mathcal{A}$, we have that

$$\mathbb{E}\left[ \sum_{t=1}^{T} \tilde{\ell}_t(E_t) \right] - \min_{e \in \mathcal{E}} \sum_{t=1}^{T} \tilde{\ell}_t(e) \leq \mathrm{R}_{\mathcal{A}}(T, (NT)^{2S}).$$

By definition of the meta loss vectors, we have that

$$\mathbb{E}\left[ \sum_{t=1}^{T} \ell_t(J_t) \right] - \min_{e \in \mathcal{E}} \sum_{t=1}^{T} \ell_t(e(t)) \leq \mathrm{R}_{\mathcal{A}}(T, (NT)^{2S}).$$

Let $j_{1:T}^{\star} \in [N]^T$ be the minimizer of $\sum_{t=1}^{T} \ell_t(j_t)$ such that $c^{\star} := \sum_{t=1}^{T-1} \mathbb{1}\{j_{t+1}^{\star} \neq j_t^{\star}\} \leq S$. Let $(t_1^{\star}, t_2^{\star}, \ldots, t_{c^{\star}}^{\star})$ be the time points where the switches in $j_{1:T}^{\star}$ occur. Observe that there exists an expert $e^{\star} \in \mathcal{E}$ which plays expert $j_i^{\star}$ between time points $t_{i-1}^{\star}$ and $t_i^{\star}$ for every $i \in [c^{\star}]$. Accordingly, we have that $e^{\star}(t) = j_t^{\star}$ for all $t \in [T]$ and

$$\mathbb{E}\left[ \sum_{t=1}^{T} \ell_t(J_t) \right] - \sum_{t=1}^{T} \ell_t(j_t^{\star}) \leq \mathrm{R}_{\mathcal{A}}^{\mathrm{obl}}(T, (NT)^{2S}),$$

completing the proof of Claim (1).

We now prove Claim (2). Consider two neighboring sequences of loss functions $\ell_1, \ldots, \ell_T$ and $\ell_1', \ldots, \ell_T'$ which differ at exactly one time point $t'$. Consider the sequence of meta loss vectors $\tilde{\ell}_1, \ldots, \tilde{\ell}_T$ and $\tilde{\ell}_1', \ldots, \tilde{\ell}_T'$ that $\mathcal{B}$ would construct and pass to $\mathcal{A}$ had it been run on $\ell_1, \ldots, \ell_T$ and $\ell_1', \ldots, \ell_T'$ respectively. Observe that $\tilde{\ell}_1, \ldots, \tilde{\ell}_T$ and $\tilde{\ell}_1', \ldots, \tilde{\ell}_T'$ are also neighboring sequence of loss functions that differ only at time point $t'$. Hence, the outputs of $\mathcal{A}$ when run $\tilde{\ell}_1, \ldots, \tilde{\ell}_T$ and $\tilde{\ell}_1', \ldots, \tilde{\ell}_T'$ are $(\varepsilon, \delta)$-indistinguishable. The proof is complete after noting that the outputs of $\mathcal{B}$ is the result of post-processing the output of $\mathcal{A}$ since the outputs of the meta-experts are fixed, and do not depend on the observed loss sequence. $\square$

We now provide concrete upper bounds on the expected dynamic regret under oblivious adversaries by instantiating Theorem 4.1 with existing private algorithms from literature. First, we recall the regret guarantee of the private online learning algorithm from Asi et al. (2024).

**Proposition 4.2** (Upper bound on Expected Regret for Oblivious Adversaries (Asi et al., 2024)). *Fix $\varepsilon, \delta \in (0, 1)$.*

*There exists an $(\varepsilon, \delta)$-differentially private algorithm $\mathcal{A}$ such that*

$$\mathrm{R}^{\mathrm{obl}}_{\mathcal{A}}(T, N, S) = O\left( \sqrt{T \log N} + \frac{T^{1/3} \log(T/\delta) \log N}{\varepsilon^{2/3}} \right).$$

Instantiating Theorem 4.1 with the algorithm guaranteed by Proposition 4.2 then gives the following Corollary.

**Corollary 4.3** (Upper bounds for Expected Dynamic Regret for Oblivious Adversaries)**.** *Fix $\varepsilon, \delta \in (0, 1)$. There exists an $(\varepsilon, \delta)$-differentially private algorithm $\mathcal{B}$ such that*

$$\mathrm{DR}^{\mathrm{obl}}_{\mathcal{B}}(T, N, S) = O\Big( \sqrt{ST \log(NT)}$$
$$+ \frac{ST^{1/3} \log(T/\delta) \log(NT)}{\varepsilon^{2/3}} \Big).$$

We highlight that Corollary 4.3 provides the first known upper bounds on expected dynamic regret under oblivious adversaries and differential privacy. Unfortunately, unlike our algorithms in Sections 3 and 5, the algorithm obtaining the upper bound in Corollary 4.3 is not efficient as it requires constructing a set of experts that is exponential in the time horizon. In Section 5, we give an efficient $\varepsilon$-differentially private algorithm whose expected dynamic regret is at most $O\left( \frac{\sqrt{ST} \log^{1.5}(NT)}{\varepsilon} + \frac{S \log(NT)}{\varepsilon} \right)$ under an adaptive adversary. Clearly, the same upper bound holds for oblivious adversaries. However, this upper bound is weaker than the one we get in Corollary 4.3.

We end this section by noting that efficient dynamic regret minimizing algorithms for oblivious adversaries do exist if one does not care about privacy. Unfortunately, unlike for static regret, privatizing existing dynamic regret minimizing algorithms is not as straightforward. As an example, Lu & Zhang (2019) give an efficient non-private algorithm for minimizing dynamic regret by projecting the distributions over experts obtained after the multiplicative weights update into a clipped simplex. This ensures that the probability of any playing any particular expert is sufficiently lower bounded. Unfortunately, this clipping operation is challenging under privacy constraints as now we have to privatize each gradient separately instead of privatizing the sum of gradients via the Binary Tree Mechanism (Dwork et al., 2010a). We leave whether one can achieve the upper bound in Corollary 4.3 via an efficient algorithm as an open question.

## 5. Dynamic Regret for Adaptive Adversaries

Under expected (static) regret, Asi et al. (2023b) prove a separation between oblivious and adaptive adversaries. In particular, for every $\varepsilon \leq \frac{1}{\sqrt{T}}$, there exists a $(\varepsilon, \delta)$ differentially private online learning algorithm whose expected

regret under oblivious adversaries is sublinear in the time horizon $T$. However, this is not the case under adaptive adversaries: for any $\varepsilon \leq \frac{1}{\sqrt{T}}$, every $(\varepsilon, \delta)$-differentially private online learning algorithm must suffer expected regret which grows linearly with $T$. In this section, we prove a qualitatively similar, but quantitatively stronger separation between private expected regret minimization under oblivious dynamic adversaries and adaptive dynamic adversaries.

### 5.1. Lower Bounds for Adaptive Adversaries

Our first result is a lower bound which roughly shows that when $\varepsilon \in o(\sqrt{\frac{S}{T}})$, sublinear expected dynamic regret is not possible under adaptive adversaries. Our lower bound construction builds upon the lower bound construction by Asi et al. (2023b) for expected (static) regret under adaptive adversaries. Namely, if there are $S$ switches, then our lower bounds follows by using $S$ different copies of the lower bound construction from Asi et al. (2023b) for adaptive adversaries.

**Theorem 5.1** (Lower bound on Expected Dynamic Regret for Adaptive Adversaries)**.** *Let $S \geq 0$, $T$ be sufficiently large, and $N \geq \frac{2T}{S}$. Let $\varepsilon \leq 1$ and $\delta \leq \frac{(S+1)^3}{T^3}$. If $\mathcal{A}$ is $(\varepsilon, \delta)$-differentially private, then*

$$\mathrm{DR}^{\mathrm{adap}}_{\mathcal{A}}(T, N, S) = \Omega\left( \min\left( T, \frac{S}{(\varepsilon \log \frac{T}{S+1})^2} \right) \right).$$

Theorem 5.1 roughly implies that when $\varepsilon \leq \sqrt{\frac{S}{T}}$, every $(\varepsilon, \delta)$-differentially private online learner must suffer expected dynamic regret $\Omega(T)$ under an adaptive adversary. This is in stark contrast to Theorem C.2 which shows that sublinear expected dynamic regret is still possible when $\varepsilon \leq \sqrt{\frac{S}{T}}$ under an oblivious adversary. The proof of Theorem 5.1 can be found in Appendix D.

### 5.2. Upper bounds for Adaptive Adversaries

Our second result is an upper bound which shows that sublinear expected dynamic regret under an adaptive adversary is possible as long as $\varepsilon = \omega(\sqrt{\frac{S}{T}})$. To do so, we modify an existing efficient (non-private) algorithm for regret minimization under adaptive dynamic adversaries. Namely, we design a private version of Algorithm 2 from Lu & Zhang (2019) by adding independent Laplace noise to the loss vectors before using them to update the distribution over the experts. For completeness sake, we include this modified algorithm below. Let $\tilde{\Delta}_N := \{w \in \Delta_N : \min_j w(j) \geq \frac{S}{NT}\}$ denote the clipped simplex, $\phi : \Delta_N \to \mathbb{R}_{\leq 0}$ denote the negative Shannon entropy function $\phi(w) := \sum_{j=1}^{N} w(j) \log w(j)$, and $\mathcal{D}_\phi(\cdot || \cdot) : \Delta_N \times \Delta_N \to \mathbb{R}_{\geq 0}$ denote the Bregman divergence with respect to $\phi$, defined as $\mathcal{D}_\phi(w_1 || w_2) :=$

$$\phi(w_1) - \phi(w_2) - \langle w_1 - w_2, \nabla\phi(w_2)\rangle.$$

---

**Algorithm 3** Private Online Learner for Adaptive Adversaries

1: **Input:** $\eta > 0, \varepsilon > 0$
2: **Initialize:** $w_1(i) = \frac{1}{N}$ for $i \in [N]$
3: **for** $t = 1, 2, \ldots, T$ **do**
4:    Draw expert $J_t \sim w_t$
5:    Observe loss vector $\ell_t$ and suffer loss $\ell_t(J_t)$
6:    Sample $Z_t(i) \sim \mathsf{Laplace}(\frac{1}{\varepsilon})$ and define $\tilde{\ell}_t(i) = \ell_t(i) + Z_t(i)$ for all $i \in [N]$
7:    Update $w_{t+1} = \arg\min_{w \in \tilde{\Delta}_N}\langle w, \eta\tilde{\ell}_t\rangle + \mathcal{D}_\phi(w\|w_t)$
8: **end for**

---

**Theorem 5.2** (Upper bound on Expected Dynamic Regret for Adaptive Adversaries). *Let $\mathcal{A}$ denote Algorithm 3 when run with $\varepsilon \in (0,1)$ and $\eta = \varepsilon\sqrt{\frac{S}{T\log(NT)}}$. Then $\mathcal{A}$ is $\varepsilon$-differentially private and has*

$$\mathrm{DR}_{\mathcal{A}}^{\mathrm{adap}}(T, N, S) = O\left(\frac{\sqrt{ST}\log^{1.5}(NT)}{\varepsilon} + \frac{S\log(NT)}{\varepsilon}\right).$$

The proof of Theorem 5.2 follows by combining techniques from Lu & Zhang (2019) and Agarwal & Singh (2017a), and is deferred to Appendix C.

## 6. Discussion

In this paper, we provide the first private online learning algorithms for dynamic regret minimization against three types of adversaries: switching stochastic, oblivious and adaptive. We highlight important directions of future work.

**Optimal bounds for Oblivious Adversaries.** In Section 4, we provided an upper bound of $O\left(\sqrt{ST\log(NT)} + \frac{ST^{1/3}\log(T/\delta)\log(NT)}{\epsilon^{2/3}}\right)$ on the expected dynamic regret under an oblivious adversary. We leave open whether one can prove a matching lower bound or an improved upper bound.

**Efficient algorithms for Oblivious Adversaries.** Unlike for stochastic and adaptive adversaries, our algorithm for oblivious adversary is not efficient – it constructs a set of experts that is exponential in the time horizon $T$. This motivates the design of *efficient* algorithms for dynamic regret minimization under oblivious adversaries with matching or better regret bounds. Unfortunately, our current attempts at designing efficient private algorithms against oblivious adversaries have been unsuccessful, as it is not clear how to privatize existing efficient non-private algorithms for dynamic regret. Perhaps central to the difficulty is the tension between lazy updating and obtaining small dynamic regret.

Existing techniques for obtaining private online learning algorithms under static regret rely on privatizing existing (non-private) lazy algorithms that do not switch their played experts too often (Asi et al., 2023c; 2024). Unfortunately, it is reasonable that such lazy algorithms cannot obtain good dynamic regret, as switching which expert is played is crucial to "tracking" the best expert. We will make sure to add a discussion of this in the camera-ready version.

## Impact Statement

This paper presents work whose goal is to advance the field of Machine Learning. There are many potential societal consequences of our work, none which we feel must be specifically highlighted here.

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

## A. Privacy Properties

**Lemma A.1** (Basic Composition (Corollary 3.15 in (Dwork et al., 2014))). *Let $\mathcal{X}, \mathcal{Y}_1, \mathcal{Y}_2, \ldots, \mathcal{Y}_T$ be arbitrary sets and $n \in \mathbb{N}$. Let $\mathcal{A}_1, \mathcal{A}_2, \ldots, \mathcal{A}_T$ be a sequence of randomized algorithms where $\mathcal{A}_1 : \mathcal{X}^n \to \mathcal{Y}_1$ and $\mathcal{A}_t : \mathcal{Y}_1, \ldots, \mathcal{Y}_{t-1}, \mathcal{X}^n \to \mathcal{Y}_t$ for all $t = 2, 3, \ldots, T$. If for every $t \in [T]$ and every $y_{1:t-1} \in \mathcal{Y}_1 \times \mathcal{Y}_2 \times \cdots \times \mathcal{Y}_{t-1}$, we have that $\mathcal{A}_t(y_{1:t-1}, \cdot)$ is $\varepsilon_t$-differentially private, then the overall algorithm $\mathcal{A} : \mathcal{X}^n \to \mathcal{Y}_1 \times \mathcal{Y}_2 \times \cdots \times \mathcal{Y}_T$, defined as*

$$\mathcal{A}(x_{1:n}) = \Big( \mathcal{A}_1(x_{1:n}), \mathcal{A}_2(\mathcal{A}_1(x_{1:n}), x_{1:n}), \ldots, \mathcal{A}_T(\mathcal{A}_1(x_{1:n}), \mathcal{A}_2(\mathcal{A}_1(x_{1:n}), x_{1:n}), \ldots, x_{1:n}) \Big),$$

*satisfies $\varepsilon T$-differential privacy.*

**Lemma A.2** (Basic Composition (Corollary 3.15 in (Dwork et al., 2014))). *Let $\mathcal{X}, \mathcal{Y}_1, \mathcal{Y}_2, \ldots, \mathcal{Y}_T$ be arbitrary sets and $n \in \mathbb{N}$. Let $\mathcal{A}_1, \mathcal{A}_2, \ldots, \mathcal{A}_T$ be a sequence of randomized algorithms where $\mathcal{A}_1 : \mathcal{X}^n \to \mathcal{Y}_1$ and $\mathcal{A}_t : \mathcal{Y}_1, \ldots, \mathcal{Y}_{t-1}, \mathcal{X}^n \to \mathcal{Y}_t$ for all $t = 2, 3, \ldots, T$. If for every $t \in [T]$ and every $y_{1:t-1} \in \mathcal{Y}_1 \times \mathcal{Y}_2 \times \cdots \times \mathcal{Y}_{t-1}$, we have that $\mathcal{A}_t(y_{1:t-1}, \cdot)$ is $\varepsilon_t$-differentially private, then the overall algorithm $\mathcal{A} : \mathcal{X}^n \to \mathcal{Y}_1 \times \mathcal{Y}_2 \times \cdots \times \mathcal{Y}_T$, defined as*

$$\mathcal{A}(x_{1:n}) = \Big( \mathcal{A}_1(x_{1:n}), \mathcal{A}_2(\mathcal{A}_1(x_{1:n}), x_{1:n}), \ldots, \mathcal{A}_T(\mathcal{A}_1(x_{1:n}), \mathcal{A}_2(\mathcal{A}_1(x_{1:n}), x_{1:n}), \ldots, x_{1:n}) \Big),$$

*satisfies $\varepsilon T$-differential privacy.*

**Lemma A.3** (Advanced Composition (Dwork et al., 2010b; Kairouz et al., 2015)). *Let $\mathcal{X}, \mathcal{Y}_1, \mathcal{Y}_2, \ldots, \mathcal{Y}_T$ be arbitrary sets and $n \in \mathbb{N}$. Let $\mathcal{A}_1, \mathcal{A}_2, \ldots, \mathcal{A}_T$ be a sequence of randomized algorithms where $\mathcal{A}_1 : \mathcal{X}^n \to \mathcal{Y}_1$ and $\mathcal{A}_t : \mathcal{Y}_1, \ldots, \mathcal{Y}_{t-1}, \mathcal{X}^n \to \mathcal{Y}_t$ for all $t = 2, 3, \ldots, T$. If for every $t \in [T]$ and every $y_{1:t-1} \in \mathcal{Y}_1 \times \mathcal{Y}_2 \times \cdots \times \mathcal{Y}_{t-1}$, we have that $\mathcal{A}_t(y_{1:t-1}, \cdot)$ is $\varepsilon_t$-differentially private, then for every $\delta' > 0$, the overall algorithm $\mathcal{A} : \mathcal{X}^n \to \mathcal{Y}_1 \times \mathcal{Y}_2 \times \cdots \times \mathcal{Y}_T$, defined as*

$$\mathcal{A}(x_{1:n}) = \Big( \mathcal{A}_1(x_{1:n}), \mathcal{A}_2(\mathcal{A}_1(x_{1:n}), x_{1:n}), \ldots, \mathcal{A}_T(\mathcal{A}_1(x_{1:n}), \mathcal{A}_2(\mathcal{A}_1(x_{1:n}), x_{1:n}), \ldots, x_{1:n}) \Big),$$

*satisfies $(\varepsilon', \delta')$-differential privacy, where*

$$\varepsilon' \le \frac{3}{2} \sum_{t=1}^{T} \varepsilon_t^2 + \sqrt{6 \sum_{t=1}^{T} \varepsilon_t^2 \log \left( \frac{1}{\delta'} \right)}.$$

Post-processing and group privacy will also be useful.

**Lemma A.4** (Post Processing (Proposition 2.1 in (Dwork et al., 2014))). *Let $\mathcal{X}, \mathcal{Y}, \mathcal{Z}$ be arbitrary sets and $n \in \mathbb{N}$. Let $\mathcal{A} : \mathcal{X}^n \to \mathcal{Y}$ and $\mathcal{B} : \mathcal{Y} \to \mathcal{Z}$ be randomized algorithms. If $\mathcal{A}$ is $(\varepsilon, \delta)$-differentially private then the composed algorithm $\mathcal{B} \circ \mathcal{A} : \mathcal{X}^n \to \mathcal{Z}$ is also $(\varepsilon, \delta)$-differentially private.*

## B. Missing Proofs for Section 3

### B.1. Proof of Theorem 3.1

The proof of Theorem 3.1 is based on the following two lemmas. The first lemma is a concentration result which shows that the average loss of each expert in each sub-interval is close to its expectation.

**Lemma B.1.** *Let $\ell_1, \ldots, \ell_T : [N] \to [0, 1]$ be sampled i.i.d. from a distribution $P$. Then with probability $1 - \beta$, for all $j \in [N]$, $t \in [T]$ and $w \in [T - t]$,*

$$\left| \sum_{i=t}^{t+w} \ell_i(j) - w \mathbb{E}_{\ell \sim P}[\ell(j)] \right| \le \sqrt{2w \log(TN/\beta)}.$$

The second lemma proves that the static regret of the algorithm with respect to the population minimizer is small.

**Lemma B.2.** *Let $\ell_1, \ldots, \ell_T : [N] \to [0, 1]$ be sampled i.i.d. from a distribution $P$. Then, with probability $1 - 3\beta$ that for all $t \in [T]$ and $w \in [T - t]$*

$$\left| \sum_{i=t}^{t+w} \ell_i(j_i) - w \min_{j \in [N]} \mathbb{E}[\ell(j)] \right| \le \frac{16 \log(NT/\beta) \log(T)}{\varepsilon} + 7\sqrt{w \log(TN/\beta)}.$$

Building on Lemma B.1 and Lemma B.2, we can now proceed to prove Theorem 3.1.

*Proof.* (of Theorem 3.1)

The privacy follows immediately from the guarantees of the report-noisy-max mechanism (Lemma 2.3): indeed, the algorithm uses the data only through the invocation of the report-noisy-max algorithm. Moreover, note that each data-point $\ell_t$ is used in a single instantiation of the report-noisy-max mechanism.

Now we proceed to prove utility. Using Lemma B.1 and Lemma B.2, we have

$$\sum_{i=t}^{t+w} \ell_i(j_i) - \min_{j \in [N]} \sum_{i=t}^{t+w} \ell_i(j) = \left( \sum_{i=t}^{t+w} \ell_i(j_i) - w \min_{j \in [N]} \mathbb{E}[\ell(j)] \right) + \left( w \min_{j \in [N]} \mathbb{E}[\ell(j)] - \min_{j \in [N]} \sum_{i=t}^{t+w} \ell_i(j) \right)$$

$$\leq \frac{16 \log(NT/\beta) \log(T)}{\varepsilon} + 7\sqrt{w \log(TN/\beta)} + \max_{j \in [N]} \left( w \mathbb{E}[\ell(j)] - \sum_{i=t}^{t+w} \ell_i(j) \right)$$

$$\leq \frac{16 \log(NT/\beta) \log(T)}{\varepsilon} + 9\sqrt{w \log(N/\beta)},$$

where the second inequality follows Lemma B.2 and the third inequality follows from Lemma B.1. $\qquad \square$

Now, it remains to prove our two lemmas. We begin with the proof of Lemma B.1.

*Proof.* (of Lemma B.1) Fix $j \in [N]$, $t \in [T]$, and $w \in [T-t]$. Since $\ell_i(j) \in [0,1]$, Hoeffding's inequality [(Duchi, 2019), Corollary 4.1.10] implies that

$$P\left( \left| \sum_{i=t}^{t+w} \ell_i(j) - w \mathbb{E}_{\ell \sim P}[\ell(j)] \right| > \sqrt{2w \log(TN/\beta)} \right) \leq \frac{\beta}{T^2 N}.$$

Taking a union bound over all $j, t, w$ proves the claim. $\qquad \square$

Finally, we prove Lemma B.2.

*Proof.* (of Lemma B.2) First, concentration of Laplace random variables [(Dwork & Roth, 2014), Fact 3.7] implies that $|Z_t(j)| \leq 2 \log(NT/\beta)/\varepsilon$ for all $j \in [N]$ and $t$ with probability at least $1 - \beta$. Let $j^\star = \arg\min_{j \in [N]} \mathbb{E}[\ell(j)]$. Then, Lemma B.1 implies that for all $t = 2^\ell$, we have

$$\mathbb{E}_{\ell \sim P}[\ell(j_t)] \leq \frac{1}{(t/2)} \sum_{i=t/2}^{t-1} \ell_i(j_t) + \frac{\sqrt{t \log(TN/\beta)}}{t/2}$$

$$\leq \frac{1}{(t/2)} \left( \sum_{i=t/2}^{t-1} \ell_i(j^\star) + Z_t(j^\star) - Z_t(j_t) \right) + \frac{2\sqrt{\log(TN/\beta)}}{\sqrt{t}}$$

$$\leq \frac{1}{(t/2)} \sum_{i=t/2}^{t-1} \ell_i(j^\star) + \frac{8 \log(NT/\beta)}{t\varepsilon} + \frac{2\sqrt{\log(TN/\beta)}}{\sqrt{t}}$$

$$\leq \mathbb{E}_{\ell \sim P}[\ell(j^\star)] + \frac{8 \log(NT/\beta)}{t\varepsilon} + \frac{4\sqrt{\log(TN/\beta)}}{\sqrt{t}},$$

where the second inequality follows from the definition of $j_t$ in the algorithm. Based on the lazy structure of the algorithm, this implies that for all $t \in [T]$,

$$\mathbb{E}_{\ell \sim P}[\ell(j_t)] \leq \mathbb{E}_{\ell \sim P}[\ell(j^\star)] + \frac{16 \log(NT/\beta)}{t\varepsilon} + \frac{4\sqrt{2 \log(TN/\beta)}}{\sqrt{t}}.$$

Now, we get that

$$\left| \sum_{i=t}^{t+w} \ell_i(j_i) - w \min_{j \in [N]} \mathbb{E}[\ell(j)] \right| \leq \left| \sum_{i=t}^{t+w} \ell_i(j_i) - \mathbb{E}[\ell(j_i)] \right| + \left| \sum_{i=t_0}^{t+w} \left( \mathbb{E}[\ell(j_i)] - \min_{j \in [N]} \mathbb{E}[\ell(j)] \right) \right|$$

$$\leq \left| \sum_{i=t}^{t+w} \ell_i(j_i) - \mathbb{E}[\ell(j_i)] \right| + \left| \sum_{i=t}^{t+w} \frac{16 \log(NT/\beta)}{t\varepsilon} + \frac{4\sqrt{2 \log(TN/\beta)}}{\sqrt{t}} \right|$$

$$\leq \left| \sum_{i=t}^{t+w} \ell_i(j_i) - \mathbb{E}[\ell(j_i)] \right| + \frac{16 \log(NT/\beta) \log T}{\varepsilon} + 6\sqrt{w \log(TN/\beta)}.$$

For the first term, note that for $W_i = \ell_i(j_i) - \mathbb{E}[\ell(j_i)]$, the sequence $\{W_i\}$ is a bounded difference martingale. We can use Azuma's inequality [(Duchi, 2019), Corollary 4.2.4] to get that

$$\mathbb{P}\left( \left| \sum_{i=t}^{t+w} \ell_i(j_i) - \mathbb{E}[\ell(j_i)] \right| > \sqrt{w \log(1/\beta)} \right) \leq \beta.$$

This proves the claim. □

### B.2. Proof of Theorem 3.2

For our analysis, we build on the following two lemmas. The first shows that if SVT identifies an above threshold query, then there must have been a distribution shift with high probability.

**Lemma B.3.** *Fix $i$. Then there is a distribution shift in the range $[t_i, t_{i+1}]$ with probability $1 - 2\beta$.*

*Proof.* Assume towards a contradiction that there is no distribution shift in the range $[t_i, t_{i+1}]$. Based on Theorem 3.1, we know that Algorithm 1 had near-optimal adaptive regret if the distribution does not change, that is, for all $w \leq t_{i+1} - t_i$ we have

$$\sum_{t_{i+1}-w}^{t_{i+1}} \ell_t(j_t) - \min_{j \in [N]} \sum_{t_{i+1}-w}^{t_{i+1}} \ell_t(j) \leq \mathsf{Reg}_w$$

However, as SVT identifies an above threshold query at time $t_{i+1}$, the guarantee of SVT (Lemma 2.4) imply that there is $w \leq t_{i+1} - t_i$ such that $q_w^t \geq -\alpha$, implying that

$$\sum_{t_{i+1}-w}^{t_{i+1}} \ell_t(j_t) - \min_{j \in [N]} \sum_{t_{i+1}-w}^{t_{i+1}} \ell_t(j) \geq \mathsf{Reg}_w + 1.$$

Therefore, we get a contradiction. □

Our second lemma shows that as long as SVT did not identify an above threshold query, the adaptive regret of the internal algorithm will be small.

**Lemma B.4.** *Fix $i$ and let $t_1', t_2' \in [t_i, t_{i+1} - 1]$. Letting $w = t_2' - t_1'$, we have with probability $1 - \beta$*

$$\sum_{t=t_1'}^{t_2'} \ell_t(j_t) - \min_{j \in [N]} \sum_{t=t_1'}^{t_2'} \ell_t(j) \leq \mathsf{Reg}_w + 2\alpha + 1.$$

*Proof.* Note that SVT did not identify an above threshold query at time $t_2'$; otherwise we would have $t_2' = t_{i+1}$. Therefore, setting $w = t_2' - t_1'$, the guarantees of the SVT mechanism for the query $q_w^{t_2'}$ imply that $q_w^{t_2'} \leq \alpha$ and therefore

$$\sum_{t=t_1'}^{t_2'} \ell_t(j_t) - \min_{j \in [N]} \sum_{t=t_1'}^{t_2'} \ell_t(j) \leq \mathsf{Reg}_w + 2\alpha + 1.$$

This proves the claim. □

Now we are ready to prove Theorem 3.2.

The privacy proof follows directly from the guarantees of SVT mechanism and Algorithm 1, as each user is used in the instantiation of both Algorithm 1 and SVT with parameters $\varepsilon/2$.

Now we proceed to prove utility. Based on Lemma B.3, for a shifting stochastic adversary with $S$ shifts, the algorithm restarts its internal procedure at most $\hat{S} \leq S$ times. Let $t_1, \ldots, t_{\hat{S}}$ denote these times. Note that the dynamic regret of the algorithm is

$$\max_{j_1^\star, \ldots, j_T^\star} 1\left\{\sum_{t=1}^T 1\{j_t^\star \neq j_{t+1}^\star\} \leq S\right\} \cdot \sum_{t=1}^T \ell_t(j_t) - \ell_t(j_t^\star) = \sum_{i=1}^S \sum_{t=t_i}^{t_{i+1}} \ell_t(j_t) - \ell_t(j_t^\star)$$

Let $L_i = t_{i+1} - t_i$ and $S_i = \sum_{t=t_i}^{t_{i+1}-1} 1\{j_t^\star \neq j_{t+1}^\star\}$ be the number of switches in $\{j_t^\star\}$ that the adversary makes inside the range $[t_i, t_{i+1}]$. We will prove that for all $i$ with high probability

$$\sum_{t=t_i}^{t_{i+1}} \ell_t(j_t) - \ell_t(j_t^\star) \leq 9\sqrt{(S_i+1)L_i \log(TN/\beta)} + (S_i+1)\left(\frac{16\log(NT/\beta)}{\varepsilon} + 2\alpha + 1\right). \tag{1}$$

Using inequality (1), we can now prove the theorem. Indeed, we get that the dynamic regret is upper bounded by

$$
\begin{aligned}
\sum_{t=1}^T \ell_t(j_t) - \ell_t(j_t^\star) &= \sum_{i=1}^S \sum_{t=t_i}^{t_{i+1}} \ell_t(j_t) - \ell_t(j_t^\star) \\
&\leq \sum_{i=1}^S 9\sqrt{(S_i+1)L_i \log(TN/\beta)} + (S_i+1)\left(\frac{16\log(NT/\beta)\log(T)}{\varepsilon} + 2\alpha + 1\right) \\
&\leq 9\sqrt{\sum_{i=1}^S (S_i+1)}\sqrt{\log(TN/\beta)\sum_{i=1}^S L_i} + 2S\left(\frac{16\log(NT/\beta)\log(T)}{\varepsilon} + 2\alpha + 1\right) \\
&\leq 9\sqrt{2ST\log(TN/\beta)} + 2S\left(\frac{16\log(NT/\beta)\log(T)}{\varepsilon} + 2\alpha + 1\right) \\
&\leq O\left(\sqrt{ST\log(TN/\beta)} + S\left(\frac{\log(NT/\beta)\log(T)}{\varepsilon}\right)\right),
\end{aligned}
$$

where the last inequality follows since $\alpha = \frac{16(2\log T + \log(2/\beta))}{\varepsilon}$. It remains to prove inequality (1). We fix $i = 1$ without loss of generality. Let $\bar{t}_1, \ldots, \bar{t}_{S_1} \in [t_1, t_2]$ denote the switching times of the sequence of experts $\{j_t^\star\}$ inside the range $[t_1, t_2]$, and let $j_{1,1}^\star, \ldots, j_{1,S_1}^\star$ denote the set of different experts in this range. Using Lemma B.4, we now get

$$
\begin{aligned}
\sum_{t=t_1}^{t_2} \ell_t(j_t) - \ell_t(j_t^\star) &= \sum_{s=1}^{S_1+1} \sum_{t=\bar{t}_i}^{\bar{t}_{i+1}} \ell_t(j_t) - \ell_t(j_{1,s}^\star) \\
&\leq \sum_{s=1}^{S_1+1} \sum_{t=\bar{t}_i}^{\bar{t}_{i+1}} 9\sqrt{(\bar{t}_{i+1} - \bar{t}_i)\log(TN/\beta)} + \frac{16\log(NT/\beta)\log(T)}{\varepsilon} + 2\alpha + 1 \\
&\leq 9\sqrt{S_1+1}\sqrt{(t_2-t_1)\log(TN/\beta)} + (S_1+1)\left(\frac{16\log(NT/\beta)\log(T)}{\varepsilon} + 2\alpha + 1\right).
\end{aligned}
$$

This proves that with probability $1 - \beta$ we have that the dynamic regret is upper bounded by $O\left(\sqrt{ST\log(TN/\beta)} + \frac{S\log(TN/\beta)\log(T)}{\varepsilon}\right)$. Picking $\beta = 1/T$ gives the upper bound on expectation as the dynamic regret is always bounded by $T$.

## C. Proof of Theorem 5.2

We first review a folklore result which states that for online learning algorithms which do not depend on the realizations of its past plays, expected regret under adaptive adversaries is at most the expected regret under oblivious adversaries.

**Theorem C.1** (Exercise 4.1 in Cesa-Bianchi & Lugosi (2006)). *Let* $\mathcal{A} : ([0,1]^N)^\star \to \Delta([N])$ *be any (randomized) online learning algorithm which maps a sequence of loss vectors to a distribution over experts. That is, for any sequence of loss functions* $\ell_1, \ldots, \ell_T$, *the prediction of* $\mathcal{A}$ *on round* $t \in [T]$ *only depends on the loss vectors* $\ell_1, \ldots, \ell_{t-1}$. *Then,*

$$\mathrm{R}_\mathcal{A}^{\mathrm{adap}}(T, N) \leq \mathrm{R}_\mathcal{A}^{\mathrm{obl}}(T, N).$$

As a consequence of Theorem C.1 and the fact that distributions constructed by Algorithm 3 do not depend on the realizations of it past plays, it is without loss of generality to consider an oblivious dynamic adversary.

To that end, we first prove the following result.

**Theorem C.2.** *Fix a sequence of loss functions* $\ell_1, \ldots, \ell_T$. *Algorithm 3, when run with* $\varepsilon, \eta > 0$ *is* $\varepsilon$-*differentially private and satisfies*

$$\mathbb{E}\left[\sum_{t=1}^T \ell_t(J_t) - \min_{j_{1:T} \in \mathcal{C}(T,S)} \sum_{t=1}^T \ell_t(j_t)\right] \leq O\left(\frac{\eta \log^2(NT)}{\varepsilon^2}T + \frac{S\log(NT)}{\eta} + \frac{S\log(NT)}{\varepsilon}\right).$$

The following lemma about Laplace vectors will be useful.

**Lemma C.3** (Norms of Laplace Vectors (Fact C.1 in (Agarwal & Singh, 2017b))). *If* $Z_1, \ldots, Z_T \sim (\mathrm{Lap}(\lambda))^N$, *then*

$$\mathbb{P}(\exists t \in [T] : ||Z_t||_\infty^2 \geq 10\lambda^2 \log^2(NT)) \leq \frac{1}{T}$$

We are now equipped to prove Theorem C.2. Our proof of utility will closely follow Theorem 1 in Lu & Zhang (2019) but account for the fact that the loss vectors used to update the algorithm can now contain large negative entries.

*Proof.* (of utility in Theorem C.2). Let $\ell_1, \ldots, \ell_T$ be the sequence of losses chosen by the oblivious adversary. Let $Z_1, \ldots, Z_T$ be the sequence of Laplace random vectors sampled in Line 6 of Algorithm 3. Observe that $Z_t \sim (\mathrm{Laplace}(\frac{1}{\varepsilon}))^N$ for all $t \in [T]$. Let $F$ be the event that there exists a $t \in [T]$ such that $||Z_t||_\infty^2 \geq \frac{10\log^2(NT)}{\varepsilon^2}$. Then, by Lemma C.3, we know that $\mathbb{P}(F) \leq \frac{1}{T}$.

Fix any sequence of experts $j_{1:T} \in \mathcal{C}(T, S)$. Observe that

$$\mathbb{E}\left[\sum_{t=1}^T \ell_t(J_t) - \sum_{t=1}^T \ell_t(j_t)|F\right] \leq T.$$

Hence, we have that

$$\mathbb{E}\left[\sum_{t=1}^T \ell_t(J_t) - \sum_{t=1}^T \ell_t(j_t)\right] \leq \mathbb{E}\left[\sum_{t=1}^T \ell_t(J_t) - \sum_{t=1}^T \ell_t(j_t)|F^c\right] + 1.$$

Using the facts that $\mathbb{E}[Z_t|F^c] = 0$, the randomness in $Z_t$ is independent of that of Algorithm 3, and $J_t$, being a function of only the past loss vectors $\ell_1, \ldots, \ell_{t-1}$, is independent of $Z_t$, we have that

$$\mathbb{E}\left[\sum_{t=1}^T \ell_t(J_t) - \sum_{t=1}^T \ell_t(j_t)\Big|F^c\right] = \mathbb{E}\left[\sum_{t=1}^T \tilde{\ell}_t(J_t) - \sum_{t=1}^T \tilde{\ell}_t(j_t)\Big|F^c\right].$$

It now suffices to upper bound $\mathbb{E}\left[\sum_{t=1}^T \tilde{\ell}_t(J_t) - \sum_{t=1}^T \tilde{\ell}_t(j_t)|F^c\right]$. To do so, we follow the proof of Theorem 1 in Lu & Zhang (2019) and modify it where necessary to account for the fact that $||\tilde{\ell}_t||_\infty \leq \frac{4\log NT}{\varepsilon}$ under the event $F^c$. Let $R \subset [T-1]$ be the subset of time points such that for every $s \in R$, we have that $j_{s+1} \neq j_s$. Note that $|R| \leq S$ by definition. Split $[T]$ into $|R| + 1$ disjoint intervals $[i_1, i_2), \ldots, [i_{|R|+1}, i_{|R|+2})$ with $i_1 = 1$ and $i_{|R|+2} = T + 1$ such that for every

$s \in [|R| + 1]$, we have that $j_{i_s} = j_{i_s+1} = \cdots = j_{i_{s+1}-1}$. Fix some $s \in [|R| + 1]$, note that the expected regret in the $s$'th interval is

$$\mathbb{E}\left[\sum_{t=i_s}^{i_{s+1}-1} \langle w_t, \tilde{\ell}_t \rangle - \tilde{\ell}_t(j_t) \Big| F^c\right].$$

Define the one-hot vectors $e_1, \ldots, e_T$ such that

$$e_t(j) := \mathbb{1}\{j = j_t\}.$$

Then, we can write

$$\mathbb{E}\left[\sum_{t=i_s}^{i_{s+1}-1} \langle w_t, \tilde{\ell}_t \rangle - \tilde{\ell}_t(j_t) \Big| F^c\right] = \mathbb{E}\left[\sum_{t=i_s}^{i_{s+1}-1} \langle w_t - e_t, \tilde{\ell}_t \rangle \Big| F^c\right]. \tag{2}$$

Further define $\tilde{e}_t \in \tilde{\Delta}_N$ such that

$$\tilde{e}_t(j) := (1 - \frac{S}{T}) e_t(i) + \frac{S}{NT}.$$

Decompose the right hand side of Equation (2) as

$$\mathbb{E}\left[\sum_{t=i_s}^{i_{s+1}-1} \langle w_t - e_t, \tilde{\ell}_t \rangle \Big| F^c\right] = \mathbb{E}\left[\sum_{t=i_s}^{i_{s+1}-1} \langle w_t - \tilde{e}_t, \tilde{\ell}_t \rangle \Big| F^c\right] + \mathbb{E}\left[\sum_{t=i_s}^{i_{s+1}-1} \langle \tilde{e}_t - e_t, \tilde{\ell}_t \rangle \Big| F^c\right].$$

Using Holder's inequality and the fact that $||\tilde{\ell}_t||_\infty \leq \frac{4 \log NT}{\varepsilon}$, we can bound

$$\langle \tilde{e}_t - e_t, \tilde{\ell}_t \rangle \leq ||\tilde{e}_t - e_t||_1 ||\tilde{\ell}_t||_\infty \leq \frac{4S \log NT}{\varepsilon T}.$$

Plugging this in, we then have that

$$\mathbb{E}\left[\sum_{t=1}^{T} \tilde{\ell}_t(J_t) - \sum_{t=1}^{T} \tilde{\ell}_t(j_t) \Big| F^c\right] \leq \mathbb{E}\left[\sum_{s=1}^{|R|+1} \sum_{t=i_s}^{i_{s+1}-1} \langle w_t - \tilde{e}_t, \tilde{\ell}_t \rangle \Big| F^c\right] + \frac{4S \log NT}{\varepsilon}.$$

Decompose $\langle w_t - \tilde{e}_t, \tilde{\ell}_t \rangle$ as

$$\langle w_t - \tilde{e}_t, \tilde{\ell}_t \rangle = \langle w_t - w_{t+1}, \tilde{\ell}_t \rangle + \langle w_{t+1} - \tilde{e}_t, \tilde{\ell}_t \rangle.$$

By the proof of Lemma 3 in Lu & Zhang (2019), we have that

$$\langle w_t - w_{t+1}, \tilde{\ell}_t \rangle \leq \eta ||\tilde{\ell}_t||_{\infty,}^2.$$

Thus, under event $F^c$, we have that

$$\langle w_t - w_{t+1}, \tilde{\ell}_t \rangle \leq \frac{10\eta \log^2 NT}{\varepsilon^2}.$$

Thus, we can write

$$\mathbb{E}\left[\sum_{t=1}^{T}\tilde{\ell}_t(J_t) - \sum_{t=1}^{T}\tilde{\ell}_t(j_t)\Big|F^c\right] \leq \mathbb{E}\left[\sum_{s=1}^{|R|+1}\sum_{t=i_s}^{i_{s+1}-1}\langle w_{t+1} - \tilde{e}_t, \tilde{\ell}_t\rangle\Big|F^c\right] + \frac{10\eta T\log^2 NT}{\varepsilon^2} + \frac{4S\log NT}{\varepsilon}$$

and it suffices to bound the first term on the right hand side. We can do so by following the same steps as in Page 17-18 of Lu & Zhang (2019). Namely, under the event $F^c$, define a convex function on the clipped simplex:

$$f(w) := \langle w, \eta\tilde{\ell}_t\rangle + \mathcal{D}_\phi(w\|w_t).$$

The update rule in Algorithm 3 can now be written as:

$$w_{t+1} = \arg\min_{w\in\tilde{\Delta}_N} f(w).$$

By first order optimality, we have that

$$\langle w_{t+1} - \tilde{e}_t, \nabla f(w_{t+1})\rangle \leq 0.$$

This gives us that

$$\eta\langle w_{t+1} - \tilde{e}_t, \tilde{\ell}_t\rangle \leq \langle \tilde{e}_t - w_{t+1}, \nabla\phi(w_{t+1}) - \nabla\phi(w_t)\rangle.$$

Thus, we can write

$$\begin{aligned}\langle w_{t+1} - \tilde{e}_t, \tilde{\ell}_t\rangle &\leq \frac{1}{\eta}\langle \tilde{e}_t, \nabla\phi(w_{t+1}) - \nabla(w_t)\rangle - \frac{1}{\eta}\langle w_{t+1}, \nabla\phi(w_{t+1}) - \nabla\phi(w_t)\rangle \\ &= \frac{1}{\eta}\langle \tilde{e}_t, \nabla\phi(w_{t+1}) - \nabla\phi(w_t)\rangle - \frac{1}{\eta}\mathcal{D}_\phi(w_{t+1}\|w_t) \\ &\leq \frac{1}{\eta}\langle \tilde{e}_t, \nabla\phi(w_{t+1}) - \nabla\phi(w_t)\rangle.\end{aligned}$$

The first equality is by definition of the Bregman divergence and the last inequality is due to the fact that Bregman divergence is always non-negative. Summing over the interval, we have that

$$\begin{aligned}\mathbb{E}\left[\sum_{t=i_s}^{i_{s+1}-1}\langle w_{t+1} - \tilde{e}_t, \tilde{\ell}_t\rangle\Big|F^c\right] &\leq \sum_{t=i_s}^{i_{s+1}-1}\frac{1}{\eta}\langle \tilde{e}_t, \nabla\phi(w_{t+1}) - \nabla\phi(w_t)\rangle \\ &= \frac{1}{\eta}\langle \tilde{e}_{i_s}, \nabla\phi(w_{i_{s+1}}) - \nabla\phi(w_{i_s})\rangle \\ &= \frac{1}{\eta}\sum_{j=1}^{N}\tilde{e}_{i_s}(j)\log\frac{w_{i_{s+1}}(j)}{w_{i_s}(j)} \\ &\leq \frac{1}{\eta}\sum_{j=1}^{N}\tilde{e}_{i_s}(j)\log\frac{NT}{S} \\ &\leq \frac{\log NT}{\eta}.\end{aligned}$$

Thus, overall, we have that

$$\mathbb{E}\left[\sum_{t=1}^{T}\tilde{\ell}_t(J_t) - \sum_{t=1}^{T}\tilde{\ell}_t(j_t)\Big|F^c\right] \le \frac{(|R|+1)\log NT}{\eta} + \frac{10\eta T\log^2 NT}{\varepsilon^2} + \frac{4S\log NT}{\varepsilon}$$

$$\le \frac{2S\log NT}{\eta} + \frac{10\eta T\log^2 NT}{\varepsilon^2} + \frac{4S\log NT}{\varepsilon}.$$

To complete the proof, recall that

$$\mathbb{E}\left[\sum_{t=1}^{T}\ell_t(J_t) - \sum_{t=1}^{T}\ell_t(j_t)\right] \le \mathbb{E}\left[\sum_{t=1}^{T}\tilde{\ell}_t(J_t) - \sum_{t=1}^{T}\tilde{\ell}_t(j_t)\Big|F^c\right] + 1$$

and hence

$$\mathbb{E}\left[\sum_{t=1}^{T}\ell_t(J_t) - \sum_{t=1}^{T}\ell_t(j_t)\right] \le \frac{2S\log NT}{\eta} + \frac{10\eta T\log^2 NT}{\varepsilon^2} + \frac{4S\log NT}{\varepsilon} + 1.$$

The upper bound in Theorem 5.2 follows after picking $\eta = \varepsilon\sqrt{\frac{S}{T\log(NT)}}$. $\qquad\square$

*Proof.* (of privacy in Theorem C.2) Let $\ell_1, \ldots, \ell_T$ and $\ell'_1, \ldots, \ell'_T$ be two sequences of neighboring loss vectors. Suppose they differ at time step $t'$. Observe that the plays of Algorithm 3 are a post-processing of the noisy losses $\tilde{\ell}_1, \ldots, \tilde{\ell}_T$ and $\tilde{\ell}'_1, \ldots, \tilde{\ell}'_T$. The distribution of the noisy losses between the two neighboring sequences remained unchanged except on round $t'$. However, since each loss vector has sensitivity 1, by the Laplace mechanism and Lemma 2.2 , we know that the output distribution for the noisy loss vector in round $t'$ is $\varepsilon$-differentially private. Thus, the overall algorithm is also $\varepsilon$-differentially private. $\qquad\square$

Theorem 5.2 in the main text follows by composing Theorem C.1 and Theorem C.2.

## D. Proof of Theorem 5.1

Before we prove Theorem 5.1, we recap the lower bound from Asi et al. (2023b).

**Proposition D.1** (Lower bound on Expected Regret for Adaptive Adversaries (Asi et al., 2023b))**.** *Let $T$ be sufficiently large and $N \ge 2T$. Let $\varepsilon \le 1$ and $\delta \le \frac{1}{T^3}$. If $\mathcal{A}$ is $(\varepsilon, \delta)$-differentially private, then*

$$R_{\mathcal{A}}^{adap}(T, N) = \Omega\left(\min\left(T, \frac{1}{(\varepsilon\log T)^2}\right)\right).$$

As mentioned in the preliminaries, an adaptive adversary for $\mathcal{A}$ for time horizon $T$ is simply a sequence of functions $f_1, f_2, \ldots, f_T$ such that at time point $t \in [T]$, the function $f_t : [N] \times [N]^{t-1} \to [0, 1]$ maps the past plays of the learning algorithm $J_1, \ldots, J_{t-1}$ to a loss vector $f_t(\cdot, J_{1:t-1}) \in [0, 1]^N$. Likewise, an online learning algorithm $\mathcal{A}$ for time horizon $T$ is a function $\mathcal{A} : ([0, 1]^N \times [N])^\star \to \Delta_N$, which at time point $t \in [T]$, takes in the past loss vectors $\ell_1, \ldots, \ell_{t-1}$, its own past plays $J_1, \ldots, J_{t-1}$, and outputs a distribution in $\Delta_N$. We will use these representations of an adaptive adversary and algorithm to prove a lower bound on expected dynamic regret for adaptive adversaries.

*Proof.* (of Theorem 5.1) Fix $S \ge 0$ and suppose without loss of generality that $S + 1$ divides $T$. Let $T' = \frac{T}{S+1}$. Let $\varepsilon \le 1$ and $\delta \le (\frac{1}{T'})^3$. Let $\mathcal{A}$ be any $(\varepsilon, \delta)$-differentially private online learning algorithm. Then, by Proposition D.1, there exists a sequence of functions $f_1^1, f_2, \ldots, f_{T'}$ such that

$$\mathbb{E}_{\mathcal{A}}\left[\sum_{t=1}^{T'} f_t(J_t, J_{1:t-1}) - \min_{j_1^\star \in [N]} \sum_{t=1}^{T'} f_t(j_1^\star, J_{1:t-1})\right] \geq \Omega\left(\min\left(T', \frac{1}{(\varepsilon \log T')^2}\right)\right),$$

where $J_t$ is the random variables denoting the prediction of $\mathcal{A}$ on round $t \in [T']$. However, now observe that we can use Proposition D.1 again starting on round $t = T' + 1$ with respect to the new internal state of $\mathcal{A}$ on round $t = T' + 1$ after fixing $J_1, \ldots, J_{T'}$. That is, by fixing $J_1, \ldots, J_{T'}$, the algorithm $\mathcal{A}$ induces a new online learning algorithm $\tilde{\mathcal{A}}$ : $([0,1]^N \times [N])^\star \to \Delta_N$ such that on input $(\ell_1, i_1), \ldots, (\ell_n, i_n) \in ([0,1]^N \times [N])^\star$ we have that

$$\tilde{\mathcal{A}}((\ell_1, i_1), \ldots, (\ell_n, i_n)) := \mathcal{A}((f_1^1(\cdot), J_1), (f_2^1(\cdot, J_1), J_2), \ldots, (f_{T'}^1(\cdot, J_{1:T'-1}), J_{T'}), (\ell_1, i_1), \ldots, (\ell_n, i_n)).$$

By post-processing, we have that $\tilde{\mathcal{A}}$ is also $(\varepsilon, \delta)$-differentially private. Note that $\tilde{\mathcal{A}}$ is random as it is a function of $J_1, \ldots, J_{T'}$. Nevertheless, Proposition D.1 guarantees the existence of a sequence of functions $\tilde{f}_1, \tilde{f}_2, \ldots, \tilde{f}_{T'}$ for $\tilde{\mathcal{A}}$ such that

$$\mathbb{E}_{\tilde{\mathcal{A}}}\left[\sum_{t=T'+1}^{2T'} \tilde{f}_{t-T'}(J_t, J_{T'+1:t-1}) - \min_{j_2^\star \in [N]} \sum_{t=T'+1}^{2T'} \tilde{f}_{t-T'}(j_2^\star, J_{T'+1:t-1})\right] \geq \Omega\left(\min\left(T', \frac{1}{(\varepsilon \log T')^2}\right)\right),$$

where now $J_{T'+1}, \ldots, J_{2T'}$ are the random variables denoting the prediction of $\tilde{\mathcal{A}}$. Recall that $\tilde{f}_1, \tilde{f}_2, \ldots, \tilde{f}_{T'}$ is a function of the realized values of $J_1, \ldots, J_{T'}$ and hence are fixed once one specifies $J_1, \ldots, J_{T'}$. Thus, we can define $f_{T'+1}, \ldots, f_{2T'}$ such that for every $t \in [T' + 1 : 2T']$ and any $j_{1:t-1} \in [N]^{t-1}$, we have that $f_t(\cdot, j_{1:t-1}) := \tilde{f}_{t-T'}(\cdot, j_{T'+1:t-1})$, where $\tilde{f}_1, \tilde{f}_2, \ldots, \tilde{f}_{T'}$ is the aforementioned strategy of the adversary when one fixes $J_1 = j_1, \ldots, J_{T'} = j_{T'}$. Now, observe that $f_{T'+1}, \ldots, f_{2T'}$ are not random and can be computed by the adversary before the game begins. Moreover, by construction, we have that

$$\mathbb{E}_{\mathcal{A}}\left[\sum_{s=1}^{2}\sum_{t=(s-1)T'+1}^{sT'} f_t(J_t, J_{1:t-1}) - \min_{j_{1:2T'}^\star \in \mathcal{C}(2T', 1)} \sum_{s=1}^{2}\sum_{t=(s-1)T'+1}^{sT'} f_t(j_t^\star, J_{1:t-1})\right] \geq \Omega\left(2\min\left(T', \frac{1}{(\varepsilon \log T')^2}\right)\right).$$

Repeating this same argument $S$ times gives a sequence of functions $f_1, f_2, \ldots, f_T$, defining the strategy of the adaptive adversary, such that

$$\mathbb{E}_{\mathcal{A}}\left[\sum_{s=1}^{S+1}\sum_{t=(s-1)T'+1}^{sT'} f_t(J_t, J_{1:t-1}) - \min_{j_{1:T}^\star \in \mathcal{C}(T, S)} \sum_{s=1}^{S+1}\sum_{t=(s-1)T'+1}^{sT'} f_t(j_t^\star, J_{1:t-1})\right] \geq \Omega\left((S+1)\min\left(T', \frac{1}{(\varepsilon \log T')^2}\right)\right)$$

$$= \Omega\left(\min\left(T, \frac{S}{(\varepsilon \log \frac{T}{S+1})^2}\right)\right).$$

This completes the proof. $\qquad\square$

