# OpenReview forum: "Tracking The Best Expert Privately"
_ICML.cc/2025/Conference — ICML 2025 poster_

### Official Review · Reviewer_Mg7a · 2025-03-12

**Overall Recommendation:** 3

**Summary:**

The authors develop differentially private algorithms for prediction with expert advice under dynamic regret (tracking the best expert) across three adversary types: stochastic with shifting distributions, oblivious, and adaptive. They achieve sub-linear regret bounds for all cases, notably providing explicit regret guarantees against shifting stochastic and oblivious adversaries.

---After Rebuttal---
---
The author fails to explain how to achieve sublinear dynamic regret in a very simple example even in the discussion stage. Consider the following 2 loss vectors when $S=1$:


(a)  $l_t(1)=1-l_t(2) = 0$ for all $t<=T/2$;   $l_t(1)=1-l_t(2) = 1$ for all $t>T/2$;

(b)  $l_t(1)=1-l_t(2) = 1$ for all $t<=T/2$;   $l_t(1)=1-l_t(2) = 0$ for all $t>T/2$;

Note that, for each of these 2 loss vectors, we can find a sequence of $\\{j^*_t\\}$ suffering $0$ "loss".


Then, since expert 1 and expert 2 are symmetric, we consider choosing expert 1 with at least 50% probability at $t=1$. Then the remaining strategy of the agent is to consider when to jump to expert 2. However, the problem is that if the agent jump to expert 2 before $t=T/2$, then the agent will inevitably suffer O(T) regret in the worst case after $t=T/2$  (i.e., in case (b)) . If the agent insists on staying in expert 1 before $t=T/2$, the agent will also suffer O(T) regret in the worst case (i.e., in cases (b) ) .

Similarly, we can get a similar conclusion if the agent chooses expert 2 with at least 50% probability at $t=1$ due to symmetry, and the worst case is case (a).

**Therefore, I have every reason to believe that the theory of this article is wrong, and hence I adjust the score to 1 and will defend this decision unless someone can point out a logical loophole in the above.**

(The authors can "edit" their own reply in the remaining rebuttal time to face this simple example directly instead of proposing an indirect answer to evade it.)

---Update 2---
---

Now, I realize that the above example is not a counter-example because the agent can switch experts more than $S$ times. I believe the writing of the paper should be largely modified. Especially, many "abuse of notation" in the problem statement would confuse the reader. In line 162, the "comparison sequence" is very confusing without pre-definition.
I will increase the score as long as I can see the detail plan for increasing the clarity of the paper.

---Update 3---
---

Thank you for the detailed response. I am looking forward to the revised version if accepted. I have increased the score to 3 as I promised, and this score is conditional on the author revising their paper as in their latest response.

**Claims And Evidence:**

The claims presented in this paper are strictly based on theoretical bounds; notably, the paper does not include any experimental validation to support these theoretical findings.

**Essential References Not Discussed:**

There is a lot of work on adversarial bandit in the literature. The author should pick a typical algorithm, such as Exp3 [1], to explain why it can or cannot be adapted to the current setting of dynamic regret with DP.

[1] Part three of Bandit Algorithms, Tor Lattimore and Csaba Szepesv ́

**Experimental Designs Or Analyses:**

Notably, the paper does not include any experimental evaluation, which I consider to be a notable weakness.

**Methods And Evaluation Criteria:**

N/A

**Other Comments Or Suggestions:**

See "Questions For Authors"

**Other Strengths And Weaknesses:**

Table 1 is particularly valuable as it clearly outlines the lower bounds for the three settings. Although the upper bounds achieved by the authors' algorithms do not yet match all of these lower bounds, the results offer important insights and guidance for future research directions.

**Questions For Authors:**

I find it difficult to gain intuitive insight into why dynamic regret is considered in this context. While static regret for adversarial bandits has been extensively studied in the literature, dynamic regret remains relatively under-explored. It would be helpful if the author could provide a clearer, more intuitive technical insight for analyzing dynamic regret, particularly highlighting scenarios where it may become suboptimal in terms of dynamic regret for the algorithms that aim at static regret.

Additionally, it would be very helpful to justify the technical insight if the authors can explain the following example: Let number of experts be 2. Let's consider a set $L=\\{0,1\\}^T$ that represents a set of loss functions. Particularly, for any $v \in L$, it implies a sequence of loss functions (or say vectors) $\mathcal{l}_t(1)=v_t$ and $\mathcal{l}_t(2)=1-v_t$ for any $t=1,\dots,T$.

Then, it looks like no matter how your agent chooses each $J_t$ at each time $t$, the expected regret must be $O(T)$  in the worst case (i.e., in one of the loss vector of the above set $L$ ) . I understand that this may not be true due to the limitation of switching times $S$; however, your definition of expected regret in Lines 145 to 152 of the second column does not explicitly depend on $S$ (particularly for $j^*$). I believe this definition needs to be revised.

Can the authors explain the above and write the intuition behind them in the Introduction section? I think this will greatly increase the readability of the paper. Especially explain the case when $S=0$ or $S=1$ in the above example.


Although my current score is negative, I am very happy to increase my score if the author can satisfactorily address these questions.

**Relation To Broader Scientific Literature:**

This paper investigates three distinct adversarial bandit frameworks with dynamic regret under differential privacy constraints: stochastic adversaries with shifting distributions, oblivious adversaries, and adaptive adversaries. By systematically addressing these settings and analyzing their dynamic regret, this work makes a comprehensive contribution to the research community.

**Theoretical Claims:**

I did not check the correctness of any proofs.

---

> ### Author Rebuttal · Authors · 2025-03-30
>
> We thank the reviewer for their comments. We address the reviewer's concerns and questions below and hope that they will reevaluate their score accordingly.
>
> > No experimental results
>
> We acknowledge the reviewer’s concern regarding the lack of experiments. However, our work is intentionally theoretical, aimed at understanding fundamental rates and limits of private learning from experts. Theoretical contributions often provide key insights that guide future empirical research. While experimental validation is certainly valuable, we believe that our results are meaningful in their own right and align with the norms of theoretical research in this area.
>
> > There is a lot of work on adversarial bandit in the literature. The author should pick a typical algorithm, such as Exp3 [1], to explain why it can or cannot be adapted to the current setting of dynamic regret with DP.
>
> We note that our work focuses on the setting of experts where we have *full-information* rather than the setting of bandits. There are many papers that study our problem (dynamic experts under *full-information* feedback) in the non-private setting which we discuss in our related work section. We will add a discussion on the challenges in privatizing these existing algorithms. We briefly mention one challenge here with regards to the modified multiplicative weights algorithm of (Lu and Zhang (2019)). Their algorithm basically applies the same update as multiplicative weights and then projects to the clipped simplex. Unfortunately, this clipping operation is challenging for privacy as now we have to privatize each gradient separately instead of privatizing the sum of gradients via the binary tree mechanism. We will discuss this more carefully in the final version.
>
> Lu, Shiyin, and Lijun Zhang. "Adaptive and efficient algorithms for tracking the best expert." arXiv preprint arXiv:1909.02187 (2019).
>
> > Motivation for dynamic regret
>
> Dynamic regret is stronger notion than static regret that compares the performance of the algorithm to a sequence of changing experts (rather than a fixed expert). This is extremely important in scenarios where there is a distribution shift which results in a new optimal expert. For example, in recommendation systems, the user's preferences might change over time.
>
> To highlight the importance of dynamic regret, lets consider the following example. Consider the setting of two experts, where the first $T/2$ losses are such that $\ell_t(1) = 0$ and $\ell_t(2)=1$, and for the last $T/2$ losses we have $\ell_t(1) = 1$ and $\ell_t(2) = 0$. For this sequence of losses, a simple algorithm that gets $0$ static regret is one that plays expert $1$ for all rounds. However, the dynamic regret of this algorithm is $T/2$ because the optimal strategy for dynamic regret would play expert $1$ for $T/2$ rounds before switching to expert $2$. For this sequence of losses, one can generalize this argument to the Multiplicative Weights (MW) algorithm, which would obtain sublinear static regret, but linear dynamic regret since MW would wait roughly $T/2 - \sqrt{T}$ rounds before switching to expert $2$. This shows a scenario where an algorithm aimed at static regret suffers suboptimal dynamic regret.
>
> > Definition of dynamic regret
>
> As for the reviewer's example and question about the definition, we first clarify that the definition of expected regret in Lines 145 to 152 is the definition of *static regret* which does not involve any switching. The definition for *dynamic regret* is provided in lines 175-181 and the constraint on the number of switches $S$ is made explicit in the definition of regret here. Note that our definition of dynamic regret  (lines 175 - 181) crucially relies on limiting the number of changes in the set of best experts we compare to ($S$ upper bounds this). Indeed, we mention in line 161 in the paper that "To make the problem tractable, we constrain the comparison sequence of experts to have at most $S$ changes". *This is crucial as otherwise we end up having linear regret, which is exactly what is happening in the reviewer's example.*
>
> When $S=0$, dynamic regret becomes equivalent to static regret since the sequence of comparison experts must consist of a single expert. When $S=1$, we allow to compare our algorithm to sequences of experts than change at most once, e.g. as in the example above, expert $1$ for the first $T/2$ rounds and expert $2$ for the last $T/2$ rounds. The larger $S$ is, the stronger is the notion of the regret. However, as $S$ becomes too large, it becomes hard to provide meaningful guarantees on the regret.
>
> We will add more clarification and intuitions regarding the parameter $S$ and its importance for the dynamic setting in the introduction.

---

> > ### Comment · Reviewer_Mg7a · 2025-04-03
> >
> > 1. Regarding the definition in Lines 145 to 152: I suggest the authors move this definition of static regret to the introduction or just under the title “2. Preliminaries.” It is very confusing to place a definition of static regret under the title “2.1. Prediction with Expert Advice and Dynamic Regret.”
> >
> > 2. Regarding $S=1$ in my example: It is still unclear how we can get sublinear regret when $S=1$. For simplicity, let’s consider the setting of an “oblivious adversary.” If your strategy is just “expert 1 for the first T/2 rounds, and expert 2 for the last T/2 rounds,” then under the losses $l_t(1) = 0$ and $l_t(2) = 1$ for all $t \leq T$, your strategy will still suffer $O(T)$ regret. I would expect a clearer explanation of how to achieve sublinear regret in my example of S=1. If I understand correctly, dynamic regret depends on the worst case of the loss vector (or say functions); and of course, different strategies may have different loss vectors as the worst case.

---

> > > ### Author Response · Authors · 2025-04-03
> > >
> > > We thank the reviewer for their response and address their comments below.
> > >
> > > 1. We thank the reviewer for the suggestion and will make sure to incorporate this change in the final version.
> > >
> > > 2. We would like to point out that we did not claim that “playing expert 1 for T/2 rounds and then switching to expert 2” is a good *algorithm* that minimizes dynamic regret for all sequences of losses. Rather, we stated that this is the optimal *strategy* for minimizing dynamic regret for just that *specific* sequence of loss functions that we considered (i.e. the first $T/2$ losses are such that $\ell_t(1) = 0$ and $\ell_t(2)=1$, and for the last $T/2$ losses we have $\ell_t(1) = 1$ and $\ell_t(2) = 0$).
> > >
> > > That said, the reviewer is correct that this particular strategy does not achieve low dynamic regret when S=1 across all loss sequences. To achieve sublinear dynamic regret when S = 1 across all loss sequences (including the reviewer's example), one has to use online learning algorithms *specifically aimed at minimizing dynamic regret*. Fortunately, there are many such algorithms (see [1, 2, 3, 4, 5, 6]) and we have referenced these works in  Section 1.2. These algorithms *modify* the standard Multiplicative Weights Algorithm to *explicitly* account for the fact that they are being evaluated against sequences of experts that can switch at most S times. At a high-level, these algorithms ensure that the probability of playing any expert never drops too low (where "low" depends on $S$ and $T$). This allows the algorithm to recover fast enough in case an expert goes from having very large loss to very small loss. So, to obtain sublinear dynamic regret when S=1, one would not use the standard Multiplicative Weights Algorithm, but use a dynamic regret minimization algorithm, like those in [1, 2, 3, 4, 5, 6], which ensures "recoverability."
> > >
> > > [1] Herbster, Mark, and Manfred K. Warmuth. "Tracking the best expert." Machine learning 32.2 (1998): 151-178.
> > >
> > > [2] Wei, Chen-Yu, Yi-Te Hong, and Chi-Jen Lu. "Tracking the best expert in non-stationary stochastic environments." Advances in neural information processing systems 29 (2016).
> > >
> > > [3] Zhang, Lijun, Shiyin Lu, and Tianbao Yang. "Minimizing dynamic regret and adaptive regret simultaneously." International Conference on Artificial Intelligence and Statistics. PMLR, 2020.
> > >
> > > [4] Herbster, Mark, and Manfred K. Warmuth. "Tracking the best linear predictor." Journal of Machine Learning Research 1.Sep (2001): 281-309.
> > >
> > > [5] Bousquet, Olivier, and Manfred K. Warmuth. "Tracking a small set of experts by mixing past posteriors." Journal of Machine Learning Research 3.Nov (2002): 363-396.
> > >
> > > [6] Zhang, Lijun, Shiyin Lu, and Zhi-Hua Zhou. "Adaptive online learning in dynamic environments." Advances in neural information processing systems 31 (2018)."
> > >
> > >
> > > ## ---RESPONSE TO UPDATE 2---
> > >
> > > The reviewer is correct that although the sequence of experts is limited to S switches, the algorithm is not. We will make this clear in the final version. Before we discuss the plan, we would like to point out that we were not trying to be evasive. As the reviewer probably realized, minimizing dyn. regret is non-trivial even when S=1. That’s why we opted to referencing existing dyn. regret minimizing algorithms, as we would not have enough space to specify these algorithms and their analysis.
> > >
> > > Detailed plan:
> > > - In the Introduction, we will include an example of how static regret minimizing algorithms can fail to achieve low dyn. regret by discussing what happens when $S$ goes from $0$ to $1$.
> > > - Within Section 2, we will make a new subsection dedicated to static regret min. and move the definition of static regret that’s in Section 2.1 to this new subsection.
> > > - Before line 162, we will define "comparison sequence of experts" and note that static regret can be written in terms of competing against the best *constant* sequence of experts.
> > > - In Section 2.1, we will discuss existing ways to minimize dyn. regret and how these differ from minimizing static regret.
> > > - In Section 2.1, we will remove all uses of “abuse of notation” and be explicit. In particular,
> > >     - In line 199, we will now define of dyn. regret under an oblivious adversary as $DR^o_A(T, N, S) := \sup_{l_1, \ldots, l_T} DR_A(f_{l_1}, \ldots, f_{l_T}, N, S)$ where $f_{l_t}$ is such that $f_{l_t}(j_t, j_{1:t-1}) := l_t(j_t).$
> > >     - In the right column on line 186, we will use a variable other than $\ell$ to define the function mapping $[N]$ and $\mathcal{Z}$ to $[0, 1]$ to avoid overloaded notation.
> > >     - In the right column on line 215-216, we will just define $A \circ Adv(z_{1:T})$ to be the sequence of random plays of the learner $A$ when interactive with the adversary given inputs $z_1,\ldots, z_T$.
> > > - In our definition of dyn. regret, we will make clear that while the sequence of experts is limited to S switches, the learner is not, and this is crucial for obtaining sublinear regret. Subsequently, we will explain how increasing $S$ makes the problem harder.

---

### Official Review · Reviewer_QJL2 · 2025-03-14

**Overall Recommendation:** 3

**Summary:**

This work studies the online private learning problem, in the setting of online prediction with experts. The main focus is the relaxed notion of dynamic regret, where the best expert of the baseline can change at most S times. The main result considers three different models of the adversary: shifting stochastic, oblivious and adaptive. For all three settings author prove upper bounds and show lower bounds in shifting stochastic and adaptive regimes. The latter indicates a separation between oblivious and adaptive online learning.

The paper uses different proof techniques: SVT-based algorithm for shifting adversary, reduction from static regret for oblivious adversary and privatization of non-private dynamic regret algorithm for adaptive adversary. For the lower bounds, authors reduce existing lower bounds for static regret.

**Claims And Evidence:**

Yes

**Essential References Not Discussed:**

Not to the best of my knowledge.

**Experimental Designs Or Analyses:**

Not applicable

**Methods And Evaluation Criteria:**

Yes

**Other Comments Or Suggestions:**

1. In Theorem 3.1, there should be additional $\log T$ factor in the first term.
2. Section 2 is not clearly written, ‘abuse of notation’ mentioned 3 times on the same page.
3. Formulation of Lemma A.3 is unclear, please rewrite.

**Other Strengths And Weaknesses:**

Strengths: The techniques used for providing upper bounds are broad and interesting. The result for the shifting adversary is nearly-optimal.

Weaknesses:
1. Upper bound for oblivious adversary is computationally inefficient.
2. It is hard to understand the tightness of the result for oblivious and adaptive adversary.

**Questions For Authors:**

1. Lower bound for oblivious adversary is the same as the shifting stochastic one. However, authors use the result of (Asi et al., 2024) to prove upper bound for oblivious adversary. This paper also contains lower bound (for certain class of algorithms). Is this lower bound extendable to dynamic regret?
2. Authors often assume 'high-dimensional' regime $T < N$ or $N > T/S$. I think there should be more discussion on what would change if $N$ is small.
3. What is $j_t$ in Lemma A.2?

**Relation To Broader Scientific Literature:**

This work extends the literature on private online learning. Results connect private online learning with static regret to dynamic regret formulation. Authors use three different techniques of establishing upper bounds in this setting.

**Theoretical Claims:**

Line 309, right column: it is claimed that when $S \ll N$, we have that $\log (N / S) = \Omega (\log N)$, which is not true generally. Consider, e.g., $N = S \log S \gg S$, but $\log N / S = \log \log S \ll \log N$.

---

> ### Author Rebuttal · Authors · 2025-03-30
>
> We thank the reviewer for their positive feedback and finding our techniques to be broad and interesting. We address the reviewer's concerns and questions below and hope that they will reevaluate their score accordingly.
>
> > Computational inefficiency
>
> This is a good question and an important direction of future work. First, we note that despite its inefficiency, this result is important on its own because *it is the first known upper bound for oblivious adversaries and it establishes a separation in the achievable rates between oblivious and adaptive adversaries*. Unfortunately, our current attempts at designing efficiency private algorithms against oblivious adversaries have been unsuccessful, as it is not clear how to privatize existing efficient non-private algorithms for dynamic regret. Perhaps central to the difficulty is the tension between lazy updating and obtaining small dynamic regret. Existing techniques for obtaining private online learning algorithms under static regret rely on privatizing existing (non-private) lazy algorithms that do not switch their played experts too often [1, 2]. Unfortunately, it is reasonable that such lazy algorithms cannot obtain good dynamic regret, as switching which expert is played is crucial to ``tracking" the best expert. We will make sure to add a discussion of this in the camera-ready version.
>
> That said, we do note that our algorithm against adaptive adversaries is efficient and also gives guarantees in the oblivious settings.
>
> [1] Asi, Hilal, et al. "Private Online Learning via Lazy Algorithms." Advances in Neural Information Processing Systems 37 (2024): 112158-112183.
>
> [2] Asi, Hilal, et al. "Private online prediction from experts: Separations and faster rates." The Thirty Sixth Annual Conference on Learning Theory. PMLR, 2023.
>
> > Tightness of the result for oblivious and adaptive adversary
>
> This is an important point of our work and we will clarify it in our final version. First, note that our bounds for oblivious adversaries are not tight as can be seen from Table 1 in the paper. As for adaptive adversaries, the bounds are tight in the following sense. Assume we ask the question of what is the best privacy (smallest $\varepsilon$) we can guarantee while being able to learn anything useful (e.g. obtain sublinear regret)? Our upper and lower bounds provide tight answer for this question: indeed, the lower bound implies that sublinear regret requires $\varepsilon = \Omega(\sqrt{S/T})$ and our upper bound (theorem 5.2) shows that the dynamic regret is sublinear whenever $\varepsilon = \Omega(\sqrt{S/T})$. This shows that our setting has $\varepsilon \ge \sqrt{S/T}$ as a fundamental threshold for learning.
>
> > Comments about lack of clarity and typos
>
> We will fix these comments in the final version. More specifically, we will add the missing $\log(T)$ factor in theorem 3.1 and rewrite Lemma A.3 more clearly. We will also revise Section 2 by removing comments regarding ``abuse of notation" and be more explicit with our definitions.
>
> >  $\log(N/S) \gg \log(N)$"
>
> The reviewer is right. When we wrote that $S \ll N$, we meant that $S$ is actually polynomial smaller than $N$ in the sense that $S = O(N^{1-\delta})$. We will clarify this in the final version.
>
> > (Q1) Extension of the lower bound of  (Asi et al., 2024)
>
> The family or algorithms used in the lower bound proof of (Asi et al. 2024) are those that are lazy, in the sense that they do not switch their played expert frequently. While this is suitable for static regret, for dynamic regret, we do want to switch experts since we are competing with a sequence of switching experts. That said, it could be extended, but it would likely not be as meaningful.
>
> > (Q2) Small $N$ regime
>
> Our focus in this work was on the high-dimensional regime where $N$ is large. However, we agree with the reviewer and believe the small $N$ regime is also of interest. Our algorithms work in that setting but are probably sub-optimal. Similarly to the static regret setting, we believe that versions on the binary tree mechanism are going to be optimal for small $N$. These algorithms usually have a worse dependence on $N$ ($N$ instead of $\log N$) but a much better dependence on $T$. We leave this question for future research.
>
>
> > (Q3) what is $j_t$ in Lemma A.2?
>
> Lemma A.2 proves certain properties about Algorithm 1 and uses notation from that algorithm ($j_t$ was defined in Algorithm 1). We will clarify this in the paper.

---

### Official Review · Reviewer_fdak · 2025-03-14

**Overall Recommendation:** 3

**Summary:**

The paper studies the dynamic regret in online learning with differential privacy. The paper considers three adversaries: shifting distributions, oblivious, and adaptive. This paper provides both lower bound and upper bound for three different adversaries. Finally, similar to static regret, This paper establishes a fundamental separation between oblivious and adaptive adversaries for the dynamic setting.

## update after rebuttal
I maintain my score.

**Claims And Evidence:**

In general, the claims are supported by sufficient evidence. The theoretical results seem sound, even though I could not understand all the proofs related to the privacy of algorithms.

**Essential References Not Discussed:**

NA

**Experimental Designs Or Analyses:**

The paper does not include experiments.

**Methods And Evaluation Criteria:**

The performance measure, the Dynamic regret and $\varepsilon$-differentially private, makes sense as it is the measure that is studied in the most similar works.

**Other Comments Or Suggestions:**

NA

**Other Strengths And Weaknesses:**

Strengths:
1. By handling shifting stochastic, oblivious, and adaptive adversaries, the authors offer a comprehensive picture, including how much privacy “costs” in each scenario.
2. The paper provides both upper bound and lower bound for dynamic regret in private online learning settings.

Weaknesses:
1. Authors may give details of the proof for differential privacy of algorithms. For example, post-processing is a crucial step in demonstrating an algorithm’s privacy. Please either include a relevant theorem in the paper or cite papers.
2. Authors may provide experiments.

**Questions For Authors:**

1. Can these results be extended to the bandit setting?
2. Is this a typo: 283-284: arg max -> arg min?

**Relation To Broader Scientific Literature:**

This paper is the first to systematically incorporate dynamic regret minimization into private online learning, bridging an important gap. Previous research on private online learning had largely focused on the static regret case.

**Theoretical Claims:**

I checked the general soundness and read a small part of the proofs, but I couldn’t read all of them given the length of the supplementary material.

---

> ### Author Rebuttal · Authors · 2025-03-30
>
> We thank the reviewer for their comments. We address the reviewer's concerns and questions below and hope that they will reevaluate their score accordingly.
>
> > Authors may give details of the proof for differential privacy of algorithms...
>
> We will make sure to include theorems about basic privacy mechanisms and their privacy/utility guarantees in the camera-ready version. That said, we do prove the privacy guarantees for all our algorithms.  For example, on lines 332-334 in the right column, we provide the proof of the privacy guarantee in Theorem 4.1. Similar privacy proofs for the other Theorems can be found in the Appendix.
>
> > Authors may provide experiments
>
> This paper is theoretical in nature as we aim at rigorously understanding fundamental limits in the context of dynamic regret minimization. While empirical validation can be valuable, our contributions stand independently as theoretical results that advance understanding in this area. Future work may explore experimental verification, but our primary goal here is to establish a solid theoretical foundation.
>
> > (Q1) Can these results be extended to the bandit setting?
>
> Yes, we do think that some of these results can be extended to the bandit setting, especially the results for stochastic and adaptive adversaries. With regards to oblivious adversaries, more care needs to be taken as one often uses unbiased estimates of the true loss under bandit feedback. This can cause issues in the privacy analysis as the sensitivity of the unbiased estimator can be large.
>
> > (Q2) Is this a typo...
>
> Yes, we thank the reviewer for catching this typo. We will make sure to fix it in the camera-ready version.

---

### Official Review · Reviewer_8ayK · 2025-03-19

**Overall Recommendation:** 3

**Summary:**

This paper studies differentially private online learning in the context of tracking the best expert, a problem where an algorithm dynamically selects from a set of experts to minimize cumulative loss over time. The authors develop differentially private algorithms for this problem under three types of adversaries: Shifting Stochastic Adversaries, where the data distribution changes up to S times;Oblivious Adversaries, which determine loss sequences independently of the algorithm. Adaptive Adversaries, which choose loss sequences based on the learner’s decisions. The paper provides upper and lower bounds on the dynamic regret of private algorithms in these settings and highlights fundamental limitations when learning privately against adaptive adversaries.

**Claims And Evidence:**

The claim presented in this paper appears to be clear and correct.

**Essential References Not Discussed:**

The paper appears to have sufficient references.

**Experimental Designs Or Analyses:**

Not applicable; the proof appears to be correct.

**Methods And Evaluation Criteria:**

The privacy and utility guarantees of the algorithms have been rigorously proven.

**Other Comments Or Suggestions:**

The paper is clearly written and well-structured. I don't have other comments.

**Other Strengths And Weaknesses:**

Strength:
* The authors provide algorithms for three types of adversaries: shifting stochastic, oblivious, and adaptive.
* The results for each setting are equipped with upper and lower bounds.


Weakness:
* The algorithm for the oblivious adversary setting is not computationally efficient, as it relies on maintaining an exponentially large set of meta-experts.

**Questions For Authors:**

The algorithm for oblivious adversaries is not computationally efficient and requires an exponential number of meta-experts. Is it possible to design an efficient algorithm that attains same-order regret bounds in this setting?

## update after rebuttal
Since my question has been answered, I will keep my positive score and vote to accept.

**Relation To Broader Scientific Literature:**

The paper contributes to differential privacy and online learning.

**Theoretical Claims:**

The proof appears to be correct.

---

> ### Author Rebuttal · Authors · 2025-03-30
>
> We thank the reviewer for their comments. We address their concerns below and hope they will reevaluate their score accordingly.
>
> > Is it possible to design an efficient algorithm that attains same-order regret bounds in this setting?
>
> This is a good question and an important direction of future work. First, we note that despite its inefficiency, this result is important on its own because *it is the first known upper bound for oblivious adversaries and it establishes a separation in the achievable rates between oblivious and adaptive adversaries*. Unfortunately, our current attempts at designing efficient private algorithms against oblivious adversaries have been unsuccessful, as it is not clear how to privatize existing efficient non-private algorithms for dynamic regret. Perhaps central to the difficulty is the tension between lazy updating and obtaining small dynamic regret. Existing techniques for obtaining private online learning algorithms under static regret rely on privatizing existing (non-private) lazy algorithms that do not switch their played experts too often [1, 2]. Unfortunately, it is reasonable that such lazy algorithms cannot obtain good dynamic regret, as switching which expert is played is crucial to ``tracking" the best expert. We will make sure to add a discussion of this in the camera-ready version.
>
> That said, we do note that our algorithm against adaptive adversaries is efficient and also gives guarantees in the oblivious settings.
>
> [1] Asi, Hilal, et al. "Private Online Learning via Lazy Algorithms." Advances in Neural Information Processing Systems 37 (2024): 112158-112183.
>
> [2] Asi, Hilal, et al. "Private online prediction from experts: Separations and faster rates." The Thirty Sixth Annual Conference on Learning Theory. PMLR, 2023.

---

### Decision · Program_Chairs · 2025-05-01

**Decision:**

Accept (poster)

**Comment:**

This paper studies the problem of differentially private bandits under dynamic regret with up to S shifts.
All reviewers agree that the paper is technically sound and is a nice addition to the literature, though no one is overly excited and the contributions are rather expected extensions than insightful new techniques.

For oblivious adversaries, the algorithm is computationally infeasible.
Another major limitation is the assumption that the number of switches is known a-piori to the learner. Most of the difficulty in the bandit literature with respect to S-switch regret has been in adapting to unknown S.